# Discrete subicular circuits control generalization of hippocampal seizures

Fan Fei[1,4], Xia Wang [1,4], Cenglin Xu [2,4] ✉, Jiaying Shi[1], Yiwei Gong [1,2], Heming Cheng [2], Nanxi Lai[1], Yeping Ruan[2], Yao Ding[3], Shuang Wang[3], Zhong Chen [1,2,3] ✉ & Yi Wang [1,2,3] ✉

Epilepsy is considered a circuit-level dysfunction associated with imbalanced excitation-inhibition, it is therapeutically necessary to identify key brain regions and related circuits in epilepsy. The subiculum is an essential participant in epileptic seizures, but the circuit mechanism underlying its role remains largely elusive. Here we deconstruct the diversity of subicular circuits in a mouse model of epilepsy. We find that excitatory subicular pyramidal neurons heterogeneously control the generalization of hippocampal seizures by projecting to different downstream regions. Notably, anterior thalamus-projecting subicular neurons bidirectionally mediate seizures, while entorhinal cortex-projecting subicular neurons act oppositely in seizure modulation. These two subpopulations are structurally and functionally dissociable. An intrinsically enhanced hyperpolarization-activated current and robust bursting intensity in anterior thalamus-projecting neurons facilitate synaptic transmission, thus contributing to the generalization of hippocampal seizures. These results demonstrate that subicular circuits have diverse roles in epilepsy, suggesting the necessity to precisely target specific subicular circuits for effective treatment of epilepsy.

Secondary generalized seizure (sGS) is a severe symptom commonly seen in temporal lobe epilepsy (TLE) that threatens ~70% of epilepsy patients[1,2] and is poorly controlled by anti-seizure drugs[3]. However, the underlying mechanism of sGS is still unclear. As epilepsy is considered a circuit-level dysfunction associated with imbalanced excitation-inhibition[4,5], it is therapeutically necessary to identify key brain regions and related circuits in TLE. The subiculum is the choke point for the output of the hippocampus[6,7] which is the common origin of epileptic seizures during TLE[8], and it sends wide projections to cortical and subcortical areas. Previously, we and other research groups have revealed that the subiculum is essential in the initiation and propagation of hippocampal seizures[9–13] and that electrical stimulation in the subiculum has a promising anti-seizure effect[14,15]. In terms of the cellular mechanism behind the dysfunctional subiculum in TLE, local

depolarized GABAergic signals[16,17] and hyperexcited long-projecting pyramidal neurons synergistically contribute to seizure activities[18–20]. However, it remains unclear how epileptic signals are transmitted from the subicular pyramidal neurons to numerous downstream regions.

Notably, the subiculum is a topologically heterogeneous structure. Recent advances have partitioned the subiculum into discrete subparts according to their disparate molecular and anatomical organizations[21–23], which motivated us to explore the subicular neural circuits involved in TLE. In the present study, we combined calcium fiber photometry, optogenetics/chemogenetics, viral-based circuit mapping, and electrophysiological recording in mouse epilepsy models, and determined the diversity of subicular circuits in TLE. We found that the SUB-ANT (anterior nucleus of thalamus) circuit bidirectionally modulated hippocampal seizures, but the SUB-EC

[1]Institute of Pharmacology and Toxicology, College of Pharmaceutical Sciences, Zhejiang University, Hangzhou, China. [2]Key Laboratory of Neuropharmacology and Translational Medicine of Zhejiang Province, School of Pharmaceutical Sciences, Zhejiang Chinese Medical University, Hangzhou, China. [3]Epilepsy Center, Second Affiliated Hospital, School of Medicine, Zhejiang University, Hangzhou, Zhejiang, China. [4]These authors contributed equally: Fan Fei, Xia Wang, Cenglin Xu. ✉e-mail: xucenglin5zz@zju.edu.cn; chenzhong@zju.edu.cn; wang-yi@zju.edu.cn

(entorhinal cortex), -MMB (mammillary bodies), and -NAc (nucleus accumbens) circuits did not. An intrinsic enhanced $I_h$-dependent burst intensity, underlying the synaptic plasticity of ANT-projecting subicular pyramidal neurons, facilitated the generalization of hippocampal seizures, which might be of therapeutic significance due to an improved understanding of the subicular neural circuit in TLE.

## Results

### Subicular pyramidal neurons are highly activated during hippocampal seizures

We first focused on functional changes of pyramidal neurons, the main type of neurons in the subiculum controlling neural transmission, during hippocampal seizures[24]. For this purpose, we expressed the genetically encoded calcium indicator GCaMP6s into subicular pyramidal neurons of wild-type mouse under the Ca/calmodulin-dependent protein kinase IIα (CaMKIIα) promoter. We subsequently implanted an optical cannula to capture calcium signaling during kindling-induced seizures[25], an animal model reflecting clinical focal seizure (FS) with sGS[26]. Immunohistochemistry analysis confirmed that GCaMP6s virus was expressed in subicular CaMKII+ neurons (90.91 ± 3.47% of CaMKII+ neurons expressed enhanced yellow fluorescent protein (YFP), Fig. 1a), suggesting that the population of calcium signaling was cell-type specific. We simultaneously recorded seizure activities and changes in fluorescence intensity and detected a significant increase of GCaMP fluorescence during both FSs (Fig. 1b) and sGSs (Fig. 1c) in the subiculum. Such seizure-induced fluorescence enhancement paralleled the intensity of epileptic behaviors and electroencephalograms (EEGs). During sGSs, the increase in fluorescence ($\Delta F/F$) was much higher than that during FSs.

Additionally, to map the neuron activation pattern during hippocampal seizures, we analyzed the expression of the immediate early gene *Fos* in the subiculum 1.5 h after seizures. There was significant c-Fos protein activation in both FS and sGS groups compared with the Sham (no seizure) group (Fig. 1d). A higher level of c-Fos expression was observed in the subiculum of mice that experienced sGSs compared to those that only experienced FSs (Fig. 1e). Immunohistochemistry showed the colocalization of c-Fos and CaMKII (over 82%, Fig. 1f), confirming that most activated subicular neurons were glutamatergic neurons. After sGSs, seizure-induced c-Fos activation was observed mainly in the deep-located subicular cells, rather than in cells in the superficial layer. No such discordance was found between the proximal and distal areas (Fig. 1g). These results indicated that subicular pyramidal neurons with sub-regional heterogeneity were highly activated during hippocampal seizures.

### Subicular pyramidal neurons with subregional heterogeneity bilaterally modulates hippocampal seizures

To investigate the causal role of subicular pyramidal neurons in hippocampal seizures, we first created non-specific lesions in the subiculum. Electrical lesioning of the subiculum showed robust anti-seizure effects on both kindling acquisition (Fig. S1a–g) and sGS expression (Fig. S1h–j), indicating that the subiculum was involved in hippocampal seizures.

We next asked whether selectively manipulating subicular pyramidal neurons influenced seizure severity. We injected AAV-CaMKIIα-hChR2-eYFP and AAV-CaMKIIα-ArchT-eGFP viruses into the subiculum of wild-type mice (CaMKIIα-ChR2 SUB and CaMKIIα-Arch SUB, respectively) to optogenetically activate (depolarize) or inactivate (hyperpolarize) these neurons[27], and injected AAV-CaMKIIα-eGFP virus for the negative control (CaMKIIα-eGFP SUB) (Fig. 1h). In vitro recordings confirmed that blue-light stimulation excited subicular pyramidal neurons (7 of 9 neurons from 3 CaMKIIα-ChR2 SUB mice, Fig. S2a, b). In hippocampal kindling model, optogenetic activation of subicular pyramidal neurons accelerated the development of behavioral seizure stage and prolonged after-discharge durations (ADD) compared with

the eGFP-light group, while inhibition delayed seizure stage and shortened ADD (Fig. 1i). The anti-seizure effect of optogenetic inhibition of the subicular pyramidal neurons was also reflected by the increasing number of stimulations needed to reach FS (stage 2) and sGS, while the pro-seizure effect of optogenetic activation of subicular pyramidal neurons was reflected by the decreasing number of stimulations needed to reach sGS (Fig. 1j, k). Typical EEGs of the hippocampal CA3 were shown in Fig. 1l. Such a pro-seizure effect was clearly observed after activation of pyramidal neurons in the deep layer of the subiculum (Fig. S3a–f), consistent with the sub-region where c-Fos was primarily activated.

In addition, we tested the influence of direct photoactivation of subicular pyramidal neurons on naive animals without hippocampal kindling. We implanted optical cannulas in CaMKIIα-ChR2 SUB mice and found that 20 Hz blue laser activation of these neurons induced seizure-like behaviors and epileptiform discharges in the hippocampal CA3 in most animals (11 of 13). With the accumulation of repetitive light stimulations, almost all the mice (10 of 11) ultimately developed sGS (Fig. S4a–c). Compared to light stimulation in the superficial subiculum group, it took less numbers of stimulation in the deep group to reached sGS (Fig. S4d, e for typical EEG recording in the CA3 and motor cortex). These results indicated that the subicular pyramidal neurons were necessary and sufficient for the generalization of hippocampal seizures.

Next, we selectively modulated subicular pyramidal neurons on sGS expression when mice were fully kindled (Fig. 1m). We found that activation of the subicular pyramidal neurons prolonged sGS durations (GSD), leaving seizure stage and ADD unaffected (Fig. 1n–p). Light stimulation itself had no influence on the seizure stage, ADD or GSD of CaMKIIα-eGFP SUB mice (Fig. S2c), suggesting such pro-seizure effect on CaMKIIα-ChR2 SUB mice was not associated with the light itself. In contrast, inactivation of subicular pyramidal neurons reduced GSD (Fig. 1q–s). However, when light stimulation was focused on the deep subiculum, it prolonged both ADD and GSD much more efficiently (Fig. S3g, h). These data suggested that subicular pyramidal neurons with subregional heterogeneity bidirectionally modulated hippocampal seizures.

### The SUB-ANT circuit bidirectionally meditates hippocampal seizures

As the subiculum is the main output region of the hippocampus, we wondered how excitatory signals passed through the subiculum during hippocampal seizures. We thus performed virus-assisted anterograde tracing and found that the subiculum projected to numerous cortical and subcortical areas, including the nucleus accumbens (NAc), medial septum (MS), lateral septum (LS), anterior nucleus of thalamus (ANT, principally in the anteroventral (AV) and anteromedial (AM) parts), laterodorsal thalamic nucleus (LD), mammillary bodies (MMB), retrosplenial granular cortex (RSG), and the deep layer of the entorhinal cortex (EC) (Fig. S5a, b). This result was consistent with several recent anatomical studies in rodents[23,28,29]. Next, we monitored c-Fos activation in these downstream areas after kindling-induced seizures. There was significant c-Fos activation in the NAc, ANT, MMB, and EC when mice underwent sGS, while few c-Fos activation was seen in the septum, LD, or RSG (Fig. S6a–h). In particular, the ANT showed dense c-Fos immunoactivity even after FS, indicating the crucial involvement of ANT cells in hippocampal seizures.

Considering that the subiculum offers numerous inputs to the ANT[30], we evaluated whether photoactivation of this pathway was necessary to aggravate seizure severity in hippocampal kindling model. We implanted an optical cannula in the ANT of CaMKIIα-ChR2 SUB mouse (CaMKIIα-ChR2SUB-ANT, Fig. 2a). Blue light stimulation in the ANT strongly promoted development of hippocampal seizures (Fig. 2b, c). When mice were fully kindled, photoactivation of SUB-ANT projections also prolonged ADD and GSD during sGS expression (Fig. 2d–f). Spectral

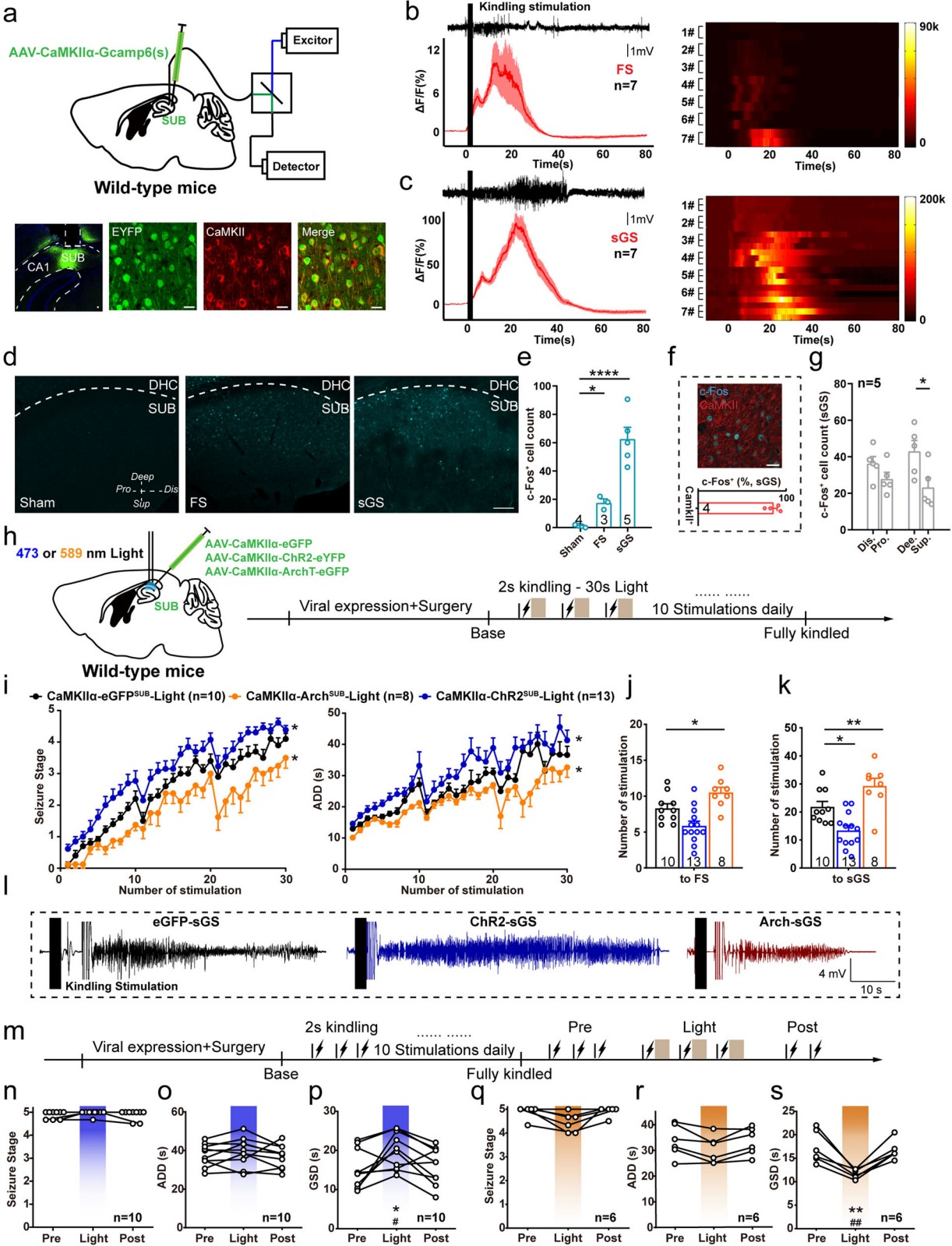

analysis revealed a significant increase in the absolute power of hippocampal EEGs during ANT terminal activation (Fig. 2g). Typical EEGs and the corresponding power spectra were shown in Fig. 2h.

Correspondingly, an optical cannula was inserted into the ANT of CaMKIIα-Arch SUB mouse to achieve terminal hyperpolarization of the SUB-ANT circuit (CaMKIIα-Arch SUB-ANT, Fig. 2i). In vitro recordings showed that yellow-light stimulation in the subiculum significantly

decreased EPSC frequency recording in the ANT pyramidal neurons (6 neurons from 3 CaMKIIα-Arch SUB mice, Fig. S7a, b), validating a successful circuit inhibition. In the kindling model, yellow light stimulation in the ANT reduced seizure stage and shortened ADD during the development of hippocampal seizures (Fig. 2j, k). Photoinhibition of this circuit also alleviated sGS expression by reducing seizure stage and decreasing ADD and GSD (Fig. 2l–n). Absolute EEG power of the

**Fig. 1 | Subicular pyramidal neurons control the generalization of hippocampal seizures. a** Top, scheme of experiments for Ca²⁺ fiber photometry of subicular pyramidal neuron during hippocampal seizures. Bottom, GCaMP6(s) expression in the subiculum (SUB), and overlap with CaMKII⁺. Scale bar, 25 μm. **b** Left, typical EEG and average calcium signals (ΔF/F) during focal seizures (FSs). Right, corresponding heatmap, each row represents one trial. **c** Left, typical EEG and average calcium signals during secondary generalized seizure (sGSs). Right, corresponding heatmap, each row represents one trial. Representative images (**d**) and quantification (**e**) of c-Fos activation pattern in the subiculum in mice with no seizure (Sham), FS, and sGS. Scale bar, 100 μm. One-way ANOVA with *post hoc* Dunnett's test, *$P < 0.05$, ****$P < 0.0001$ compared with Sham. **f** Overlap between activated c-Fos and CaMKII⁺ neurons after sGSs and quantification. Scale bar, 25 μm **g** Quantification of c-Fos in distal, proximal, deep, and superficial areas of the subiculum after sGSs. Paired t-test, *$P < 0.05$. **h** Scheme of optogenetic experiments in sGS acquisition. **i** Effects of optogenetic modulation of subicular pyramidal neurons on the development of seizure stage and after-discharge durations (ADD). Two-way repeated measures ANOVA with *post hoc* Scheffe's test, *$P < 0.05$ compared to eGFP. **j, k** Number of stimulations needed to reach FS (**j**) and sGS (**k**) during seizure development. One-way ANOVA with *post hoc* Dunnett's test, *$P < 0.05$, **$P < 0.01$. **l** Typical sGS EEGs recorded from the CA3. **m** Scheme of optogenetic experiments in sGS expression. Effects of optogenetic activation of subicular pyramidal neurons on seizure stage (**n**), ADD (**o**), and sGS durations (GSD, **p**) during sGS expression. One-way repeated measures ANOVA with *post hoc* Dunnett's test, *$P < 0.05$ compared to Pre, #$P < 0.05$ compared to Post. Effects of optogenetic hyperpolarization of subicular pyramidal neurons on seizure stage (**q**), ADD (**r**), and GSD (**s**) during sGS expression. One-way repeated measures ANOVA with *post hoc* Dunnett's test, **$P < 0.01$ compared to Pre, ##$P < 0.01$ compared to Post. The number of mice used in each group is indicated in the figure. Data are presented as means ± SEM.

CA3 was effectively reduced during ANT terminal hyperpolarization (Fig. 2o). Typical EEGs and the corresponding power spectra were shown in Fig. 2p. Together, these results indicated that the SUB-ANT circuit mediated the severity of hippocampal seizures bidirectionally.

## Inhibition of SUB-EC, MMB, and NAc terminals fails to suppress hippocampal seizures

Next, we aimed to illustrate whether other downstream projections of the SUB were involved in seizure modulation. We tested the other three regions, EC, NAc, and MMB, which also showed the activated c-Fos immunofluorescence. Optical cannulas were implanted into the EC, NAc, or MMB of CaMKIIα-Arch ᔆᵁᴮ mice to selectively inactivate (hyperpolarize) those circuit terminals (respectively CaMKIIα-Arch ᔆᵁᴮ⁻ᴱᶜ, CaMKIIα-Arch ᔆᵁᴮ⁻ᴺᴬᶜ, and CaMKIIα-Arch ᔆᵁᴮ⁻ᴹᴹᴮ, Fig. 3a, f). When the SUB-EC circuit was hyperpolarized, development of seizure stage and ADD were accelerated compared with the eGFP-light group (Fig. 3b). Moreover, hyperpolarization of the EC-projecting subicular terminal moderately prolonged ADD and GSD during sGS expression when mice were fully kindled (Fig. 3c, d). As shown in the power spectra, EEG powers were strengthened mainly during the later period when mice underwent sGSs (Fig. 3e). As comparison, hyperpolarization of NAc- and MMB-projecting subicular terminals showed no apparent effects on the development of hippocampal seizures (Fig. 3g). In addition, activation of SUB-EC, NAc, and MMB circuits also had no effects on the development of hippocampal seizures (Fig. S8a–f). Together, these results indicated a diverse influence of subicular downstream circuits on hippocampal seizures. Specifically, inhibition of the SUB-ANT circuit alleviated, while inhibition of the SUB-EC circuit aggravated, hippocampal seizures.

To verify the general phenomena of such divergence and better resemble the features of clinical epilepsy patients, we next introduced kainic acid (KA)-induced chronic TLE model. Briefly, KA (0.25 μg) was injected into the hippocampal CA1 of wild-type mice, after which recurrent spontaneous seizures typically occurred several weeks later in the chronic period[31]. To precisely suppress the neural activity of the SUB-ANT and SUB-EC circuits for long-term, we injected AAV-CaMKIIα-hM4Di-mCherry into the subiculum of wild-type mice (CaMKIIα-hM4Di ᔆᵁᴮ). This virus carries hM4Di, an engineered inhibitory Gi-coupled human muscarinic receptor, which can increase intracellular Gi signaling with the existence of clozapine-N-oxide (CNO), and thus decrease neuronal activity[32]. We also injected AAV-CaMKIIα-mCherry as comparison (CaMKIIα-mCherry ᔆᵁᴮ). Despite the fact that the subiculum was reported to experience much less cell loss during the KA chronic period than the hippocampal proper[8], we still found an obvious proportion of cellular damage in subicular neurons (approximately 39.2% neuronal loss, Fig. S9a, b). However, robust expression of mCherry immunoactivity was detected in the subiculum and its projecting downstream regions, including the ANT and EC (Fig. 4a, b). We implanted cannulas into the ANT or EC in CaMKIIα-hM4Di ᔆᵁᴮ mice for precise delivery of vehicle or CNO in projecting terminals (CaMKIIα-hM4Di ᔆᵁᴮ⁻ᴬᴺᵀ or CaMKIIα-hM4Di ᔆᵁᴮ⁻ᴱᶜ).

Local application of CNO in the ANT significantly reduced the number and time of FSs (typical EEG shown in Fig. 4c–e). In some mice, we also detected behavioral sGSs. CNO treatment also decreased the number and time of sGSs in CaMKIIα-hM4Di ᔆᵁᴮ⁻ᴬᴺᵀ mice (Fig. 4f, g). Representative EEGs and corresponding power spectra before and after CNO injection were shown in Fig. 4h. In contrast, when CNO was locally injected into the EC, the number and time of both FSs and sGSs were increased in CaMKIIα-hM4Di ᔆᵁᴮ⁻ᴱᶜ mice (Fig. 4i–m). We also observed one dead case after CNO delivery among the six mice. Besides, CNO injection in the ANT of CaMKIIα-mCherry ᔆᵁᴮ mice did not change the number and time of both FSs and sGSs (Fig. S9c–f), indicating such seizure-modulating effects were irrelevant to CNO itself. These data suggested that subicular pyramidal neurons differentially modulate hippocampal seizures via distinct downstream circuits, among which the SUB-ANT and SUB-EC had opposite contributions to hippocampal seizures.

## The ANT-projecting and EC-projecting neuron sub-populations in the subiculum are structurally and functionally heterogeneous

The above observations in both kindling and KA models led us to hypothesize that the subiculum had anatomically distinct pathways to these downstream areas, especially to the ANT and EC, and they transmitted different information during epileptic condition. To test these probabilities, we first performed retrograde tracing using cholera toxin subunit B conjugated with Alexa Fluor-555 and 647 (CTB-555 and CTB-647, Fig. 5a, b). We observed that the somatic locations of the ANT-projecting neurons and the EC-projecting neurons in the subiculum were totally segregated. Specifically, ANT-projecting neurons were almost completely deep located while EC-projecting neurons were mostly distributed in the superficial layer. Such topological distribution was consistent across the anterior-posterior axis (Fig. 5c, d). In addition, neurons projecting to the MMB and NAc were also located at different positions in the subiculum compared to the ANT-projecting neurons (Fig. S10a–d). These results indicated that most subicular neurons projecting to distinct downstream areas were not collateral projections but had structural heterogeneity. Although heterogeneity in subicular projections to cortical and subcortical regions was reported before[33,34], yet no existing studies directly distinguished the specific locations of ANT- and EC-projecting subicular pyramidal neurons.

Next, we introduced AAV-Ef1α-DIO-Axon-GCaMP6(s) into the subiculum of *CaMKIIα-cre* mice. This virus was genetically reengineered to have greater sensitivity in axon terminals, which allowed for enhanced fluorescence intensity detection at the circuit level[35]. We stereotaxically implanted optical cannulas into the ANT and EC of each mouse and monitored calcium fluorescence changes during hippocampal seizures (Fig. 5e, f). The ANT and EC terminals exhibited diverse neural activities after kindling stimulation. During FSs (Fig. 5g), the ANT terminal exhibited increased calcium signaling while the EC

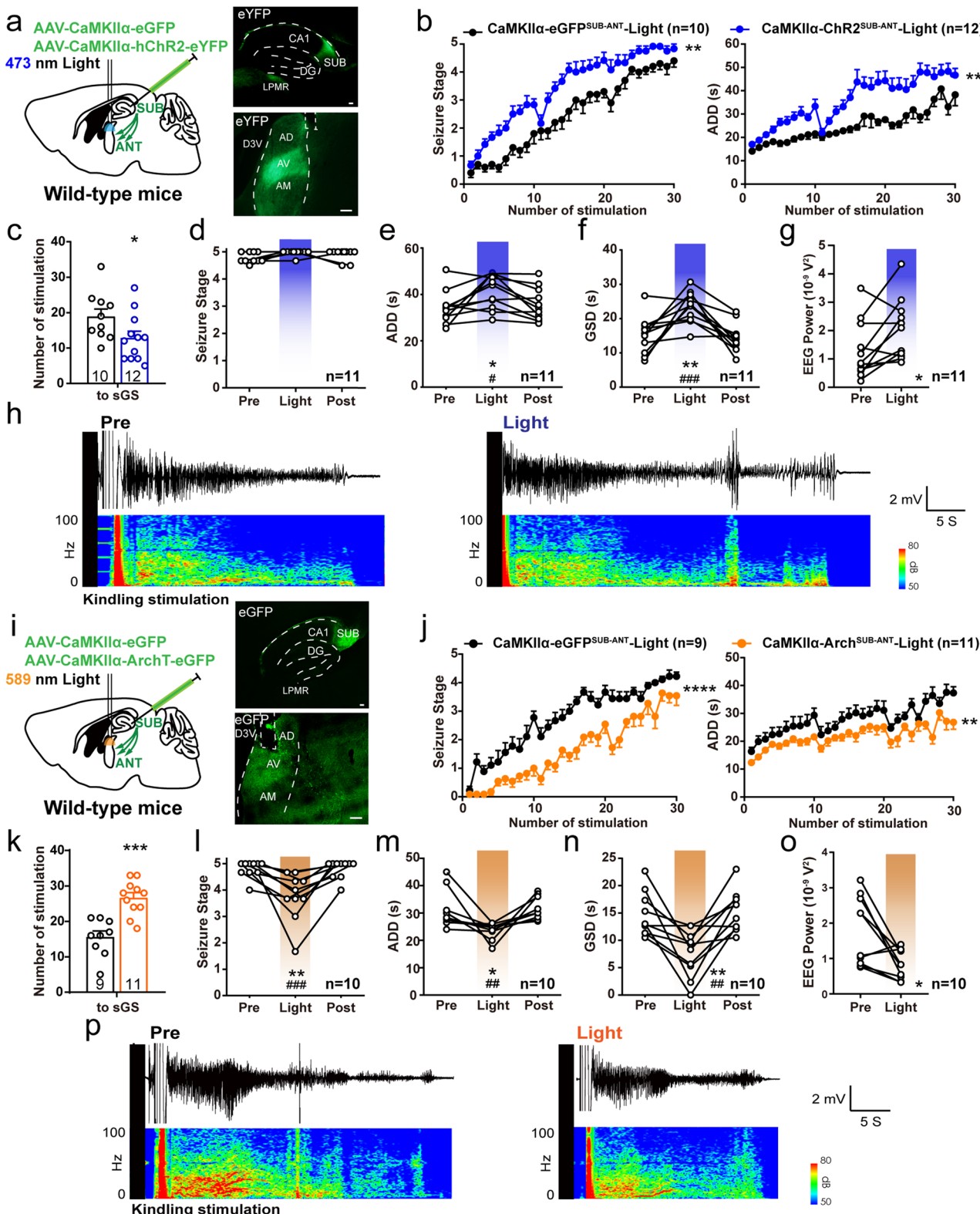

We further performed in vivo multi-unit recordings in the sub-

decreased (Fig. 5h, i). During sGSs (Fig. 5j), fluorescence in the ANT terminal showed a delayed increase with seizure development (Fig. 5k, l); while in the EC terminal, fluorescence temporarily increased (corresponding to behavioral latency to GS) but then decreased even below the baseline, which lasted long after sGS termination (Fig. 5m, n). These results illustrated that the functional response of these two pyramidal neuron sub-populations to hippocampal seizures were totally separated.

iculum (Fig. 6a) and defined the location of recording units in the deep (SUBa, the main position of the ANT-projecting neurons) or superficial areas of the subiculum (SUBb, the main position of the EC-projecting neurons) with post verification (Fig. S11). Although units in the SUBa and SUBb were not discriminative in their average firing rates, units in the SUBa had a more robust bursting firing ability (higher number of bursts per min, Fig. 6b–d). To more precisely identify and characterize

**Fig. 2 | SUB-ANT glutamatergic circuit bidirectionally mediates hippocampal seizures. a** Scheme of experiments and a representative sagittal image for ChR2-eYFP expression in the subiculum (SUB) and light stimulation in the anterior nucleus of thalamus (ANT). LPMR, lateral posterior thalamic nucleus, mediorostral part. AD, anterodorsal thalamic nucleus; AV, anteroventral thalamic nucleus; AM, anteromedial thalamic nucleus; D3V, dorsal 3rd ventricle. Scale bar, 100 μm. **b** Effects of optogenetic activation of SUB-ANT circuit on the development of seizure stage and after-discharge durations (ADD). Two-way repeated measures ANOVA, **$P < 0.01$, ***$P < 0.001$. **c** Number of stimulations needed to reach secondary generalized seizure (sGS). Unpaired $t$-test, *$P < 0.05$. Effects of optogenetic activation of SUB-ANT circuit on seizure stage (**d**), ADD (**e**), and GS durations (GSD, **f**) during sGS expression. Friedman with *post hoc* Dunn's test for seizure stage. One-way repeated measures ANOVA with *post hoc* Dunnett's test for ADD and GSD, *$P < 0.05$, **$P < 0.01$ compared to Pre, #$P < 0.05$, ###$P < 0.001$ compared to Post. **g** Spectral analysis of sGS EEG power (0–100 Hz) of the CA3. Paired $t$-test, *$P < 0.05$.

**h** Representative CA3 EEGs and power spectrogram during sGSs. **i** Scheme of experiments and a representative sagittal image for Arch-eGFP expression in the subiculum and light stimulation in the ANT. Scale bar, 100 μm. **j** Effects of optogenetic inhibition of SUB-ANT circuit on the development of seizure stage and ADD. Two-way repeated measures ANOVA, **$P < 0.01$, ****$P < 0.0001$. **k** Number of stimulations needed to reach sGS. Unpaired $t$-test, ***$P < 0.001$. Effects of optogenetic inhibition of SUB-ANT circuit on seizure stage (**l**), ADD (**m**), and GSD (**n**) during sGS expression. Friedman with *post hoc* Dunn's test for seizure stage. **$P < 0.01$ compared to Pre, ###$P < 0.001$ compared to Post. One-way repeated measures ANOVA with *post hoc* Dunnett's test for ADD and GSD. *$P < 0.05$, **$P < 0.01$ compared to Pre, ## $P < 0.01$ compared to Post. **o** Spectral analysis of sGS EEG power (0–100 Hz) of the CA3. Paired $t$-test, *$P < 0.05$. **p** Representative CA3 EEGs and power spectrogram during sGSs. The number of mice used in each group is indicated in figure. Data are presented as means ± SEM.

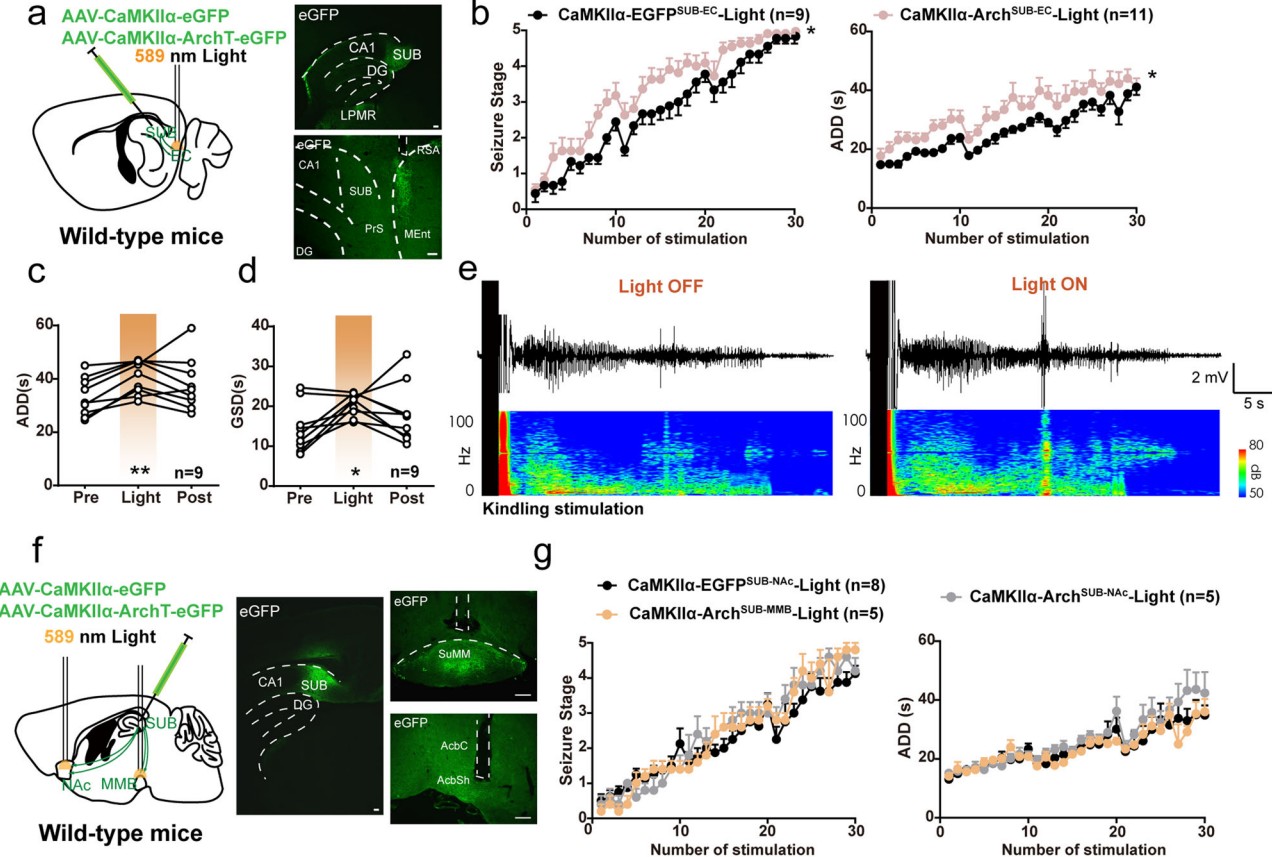

**Fig. 3 | SUB-EC, SUB-MMB, and SUB-NAc circuits function differently in modulating hippocampal seizures. a** Scheme of experiments and a representative sagittal image for Arch-eGFP expression in the subiculum (SUB) and light stimulation in the entorhinal cortex (EC). Scale bar, 100 μm. **b** Effects of optogenetic inhibition of SUB-EC circuit on the development of seizure stage and after-discharge durations (ADD). Two-way repeated measures ANOVA, *$P < 0.05$. **c, d** Effects of optogenetic inhibition of SUB-EC circuit on the ADD (**c**) and generalized seizure durations (GSD, **d**) during sGS expression. One-way repeated

measures ANOVA with *post hoc* Dunnett's test, *$P < 0.05$ compared to Pre. **$P < 0.01$ compared to Pre. **e** Representative CA3 EEGs and power spectrogram during sGSs. **f** Scheme of experiments and a representative coronal image for Arch-eGFP expression in the subiculum (SUB) and light stimulation in the the mamillary bodies (MMB) or nucleus accumbens (NAc). Scale bar, 100 μm. **g** Effects of optogenetic inhibition of SUB-MMB and SUB-NAc circuits on the development of seizure stage and ADD. The number of mice used in each group is indicated in figure. Data are presented as means ± SEM.

circuit-specific subicular neurons, we next introduced CTB-based slice electrophysiology (Fig. 6e). We injected CTB-555 in the ANT and performed patch-clamp recordings in the subiculum. Almost all ANT-projecting neurons were distributed in the area we defined as SUBa. In total, we recorded 52 pyramidal neurons from 15 mice, among which 25 fluoresced red. The basic electrophysiological properties of these two sub-populations of pyramidal neurons showed no obvious differences, except that the ANT-projecting neurons exhibited significantly larger sags at hyperpolarizing injected currents (Table S1).

Moreover, each patched cell could fire either in regular spiking or bursting at suprathreshold (100 pA) (Fig. 6f). Sag ratios were higher in the ANT-projecting bursting neurons, but were moderate in ANT-projecting regular firing neurons and non-ANT-projecting neurons (Fig. 6g). Meanwhile, $I_h$ density was also significantly larger in the ANT-projecting neurons than non-ANT-projecting neurons (Fig. 6h). These electrophysiological properties were mediated by somatic hyperpolarization-activated cyclic nucleotide-gated cation (HCN) channel. As a result, the ANT-projecting bursting neurons showed

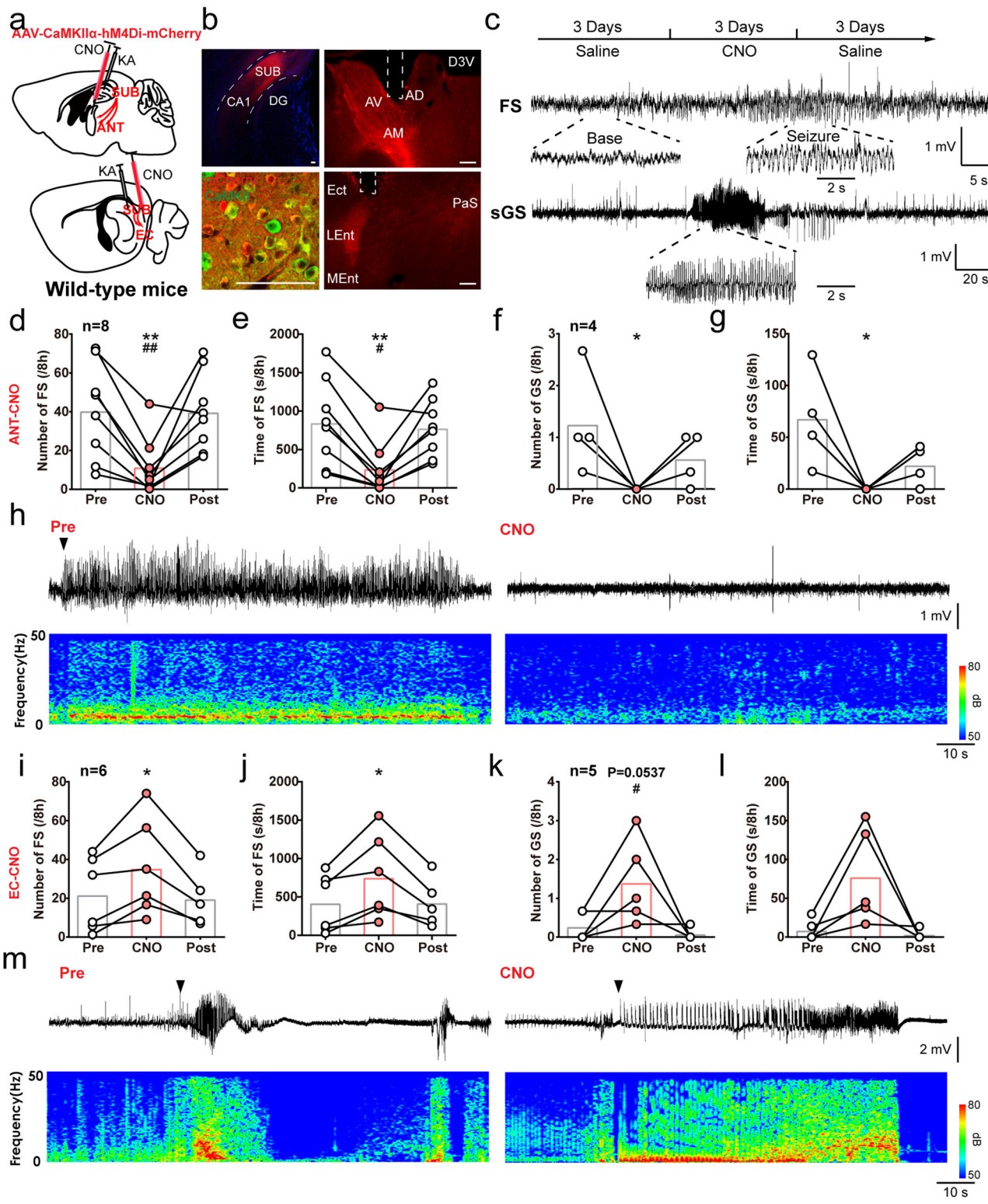

higher burst firing intensity, revealed by more spikes and shorter spike intervals in their bursting action potential (Fig. 6i, j). Such robust burst firing could be diminished using a selective HCN channel blocker, ZD7288 (Fig. 6k–m, note ZD7288 also shifted the rest membrane potential)[36], indicating that HCN channel contributed to the augmented bursting activity of ANT-projecting subicular neurons. These data demonstrated that the ANT-projecting and EC-projecting subicular neuronal sub-populations had structural and functional heterogeneity. Specifically, ANT-projecting subicular pyramidal neurons,

mainly distributed in the deep layer of the subiculum, showed enhanced HCN channel-contributed bursting intensity, and were easily activated by the hippocampal seizures.

## HCN channel-based synaptic plasticity in the SUB-ANT circuit contributes to the development of hippocampal seizures

Given that the ANT-projecting neurons showed significantly larger somatic hyperpolarized sags and bursting ability than other subicular pyramidal neurons, we sought to test whether HCN channel function in

**Fig. 4 | SUB-ANT and SUB-EC circuits have opposite effects on hippocampal seizures in KA-induced chronic TLE model. a** Scheme of experiments for chemogenetic inhibition of Sub-ANT and Sub-EC circuits in KA-induced chronic model. KA was injected in the dCA1, AAV-CamkIIα-hM4Di-mCherry virus in the SUB, and CNO in the ANT or EC. **b** Left, representative coronal images show that mCherry florescence was expressed within the SUB (top) and highly co-expressed with CaMKII (bottom). Right, representative coronal images of the cannula placement in the ANT (top) or EC (bottom). Scale bar, 75 μm. **c** Top, scheme of experiments for CNO treatment in KA-induced chronic model. Bottom, typical EEGs of spontaneous focal seizure (FS) and secondary generalized seizure (sGS). Effects of chemogenetic inhibition of SUB-ANT circuit on the number and duration of FSs (**d**, **e**) and sGSs (**f**, **g**). One-way repeated measures ANOVA with *post hoc* Dunnett's test for numbers and times of FSs, \*\**P* < 0.01 CNO compared with Pre, #*P* < 0.05, ##*P* < 0.01 CNO

compared with Post. Friedman with *post hoc* Dunn's test for numbers and times of sGSs, \**P* < 0.05 CNO compared with Pre. **h** Typical CA3 EEGs and power spectrum during one seizure in Pre-CNO and CNO injection in the ANT. The upper triangle indicates the seizure onset. Effects of chemogenetic inhibition of SUB-EC circuit on the number and duration of FSs (**i**, **j**) and sGSs (**k**, **l**). One-way repeated measures ANOVA with *post hoc* Dunnett's test for numbers and times of FSs, \**P* < 0.05 CNO compared with Pre. Friedman with *post hoc* Dunn's test for numbers and times of sGSs, *P* = 0.0537 CNO compared with Pre in numbers of sGSs, #*P* < 0.05 CNO compared with Post. **m** Typical CA3 EEGs and power spectrum of one seizure in Pre and CNO injection in the EC. The upper triangle indicates the seizure onset. The number of mice used in each group is indicated in the figure. Data are presented as means ± SEM.

these neurons directly contributes to hippocampal seizures. Therefore, we first injected ZD7288 into the subiculum of fully kindled mice. In vivo blockage of the subicular HCN channel significantly and dose-dependently reduced seizure stage, ADD, GSD, and incidence of sGSs during sGS expression (Fig. 7a, b). As isoform HCN1 is the most abundant HCN channel subtype expressed in the rodent subiculum[37], we next wondered how selectively knocking down HCN1 activity in ANT-projecting or EC-projecting subicular pyramidal neurons influenced bursting and hippocampal seizures. For this purpose, we injected retrograde Cav2-*cre* virus[38] into the ANT or EC of wild-type mice, and then injected AAV-EF1α-DIO-miR30shRNA (Hcn1)-mCherry virus into the subiculum (Fig. 7c). In vitro recordings confirmed that HCN1 knockdown significantly eliminated hyperpolarized sag and decreased bursting intensity of ANT-projecting subicular pyramidal neurons (Fig. S12a–f). In the hippocampal kindling model, selectively knocking down the expression of HCN1 in ANT-projecting subicular neurons alleviated the development of hippocampal seizures, while knocking down the HCN1 channel in the EC-projecting subicular neurons had no effect (Fig. 7d). These results indicated that HCN channel function in ANT-projecting subicular neurons contributes to hippocampal seizures. Additionally, as previous studies showed that T-type Ca²⁺ channel contributed to bursting of subicular pyramidal neurons[39], and ZD7288 could inhibit T-type Ca²⁺ current in hippocampal pyramidal neurons[40], we here confirmed this possibility (Fig. S13a–d) and further found that intra-subicular injection of T-type calcium channel blocker, TTA-P2 significantly reduced seizure stage and shortened GSD in sGS expression as well (Fig. S13e, f). These results strongly indicated that bursting firing of subicular pyramidal neurons was important in generalization of hippocampal seizures.

Further, as HCN channel-dependent bursting ability is important for efficient neurotransmission and enhanced synaptic plasticity[41,42], we asked how the SUB-ANT circuit was involved in epilepsy. We introduced single pulse test to measure the synaptic plasticity of the SUB-ANT pathway in epilepsy: apply a single electrical stimulation in the subiculum, and record a voltage response in the ANT (Fig. 7e). With the augmentation of the stimulation current, the voltage recorded in the ANT gradually increased and finally became saturated. We chose the current that induced half of the maximum voltage as the test current (Fig. 7f). Such an induced potential in the ANT was strengthened after the mice were fully kindled, suggesting enhanced synaptic transmission of the SUB-ANT circuit after mice underwent sGSs. Next, we tested the influence of blockage of subicular HCN channels or optogenetic inhibition of the SUB-ANT circuit during kindling acquisition on the strength of synaptic plasticity. Both blockage of subicular HCN channels and optogenetic inhibition of the SUB-ANT circuit during kindling suppressed the single pulse in the ANT, revealed by lower amplitude and slope of field excitatory postsynaptic potential (fEPSP) (Fig. 7g–i). In addition, direct photoinhibition of pyramidal neurons in the ANT, which are innervated by subicular output, also robustly suppressed the generalization of hippocampal seizures (Fig. S14a–f). Together, these results indicated that HCN channel-based

synaptic plasticity in the SUB-ANT circuit contributed to the development of hippocampal seizures, and inhibition of the HCN channel in this specific SUB-ANT circuit disrupted hippocampal seizures.

## Discussion

The subiculum is thought to be essential in the initiation and propagation of seizures in epilepsy, but how long-projecting subicular pyramidal neurons and related circuits recruited was not revealed before. Here, in animal models of epileptic seizures, we provided direct evidence that: (1) Seizure-induced hyperexcited subicular pyramidal neurons with subregional heterogeneity bidirectionally modulate seizures in TLE; (2) the ANT-projecting subicular circuit amplifies seizures via enhanced HCN channel-mediated synaptic transmission; and (3) the subicular projections to the EC act as a parallel but contrary pathway to attenuate seizure severity. As we previously demonstrated, selective inhibition of subicular pyramidal neurons is sufficient to reverse both phenytoin and lamotrigine resistance phenomena in animal models of epilepsy[18,19]. However, it is rather arbitrary to consider the subiculum as integral during epilepsy, as both pyramidal neurons and interneurons show diverse changes within subregions. For instance, the VGLUT1⁺ neuron is more severely degenerated in the distal subiculum compared with the proximal subiculum one month after KA-induced status epilepticus; while parvalbumin interneuron loss is far more obvious in the proximal and superficial part of the subiculum in the acute period of KA-induced seizures[43,44]. Our findings in the present study confirm that a portion of subicular pyramidal neurons, mainly locating in the deep layer of the subiculum and projecting to the ANT, are hyperexcited and contribute to seizures in TLE, and that this effect is associated with HCN channel-contributed bursting activities. These results shed light on the therapeutic significance of precise intervention targeting the subiculum during epileptic seizures.

At the circuit level, we demonstrate that the subicular afferent to the ANT largely contributes to the generalization of epileptic seizures. Selective activation of the SUB-ANT circuit promotes, while inhibition alleviates, hippocampal seizures in both sGS acquisition and sGS expression states. More broadly, in KA-induced chronic spontaneous seizures, which better represent clinical ictal events, inhibiting the SUB-ANT circuit also shows promise in terms of reducing seizure severity. This indicates that the ANT-projecting subicular circuit amplifies seizures, in accordance with our previous finding in clinical TLE with sGS patients that the thalamus may be an important structure in the generalization of seizures[45]. Despite the ANT being previously considered a key brain region involved in epilepsy and widely used as a target of deep brain stimulation therapy in clinical epilepsy[46], the underlying neural mechanism was not elucidated. Stimulation in the ANT at both low and high frequencies desynchronizes hippocampal seizures in patients and animal models[47,48]. Our results, consistent with this, showed that selectively manipulating the SUB-ANT circuit in hippocampal seizures widely influenced absolute power in EEG spectra, suggesting that inhibiting the SUB-ANT circuit may desynchronize hippocampal seizures. An important question is how inhibition of ANT

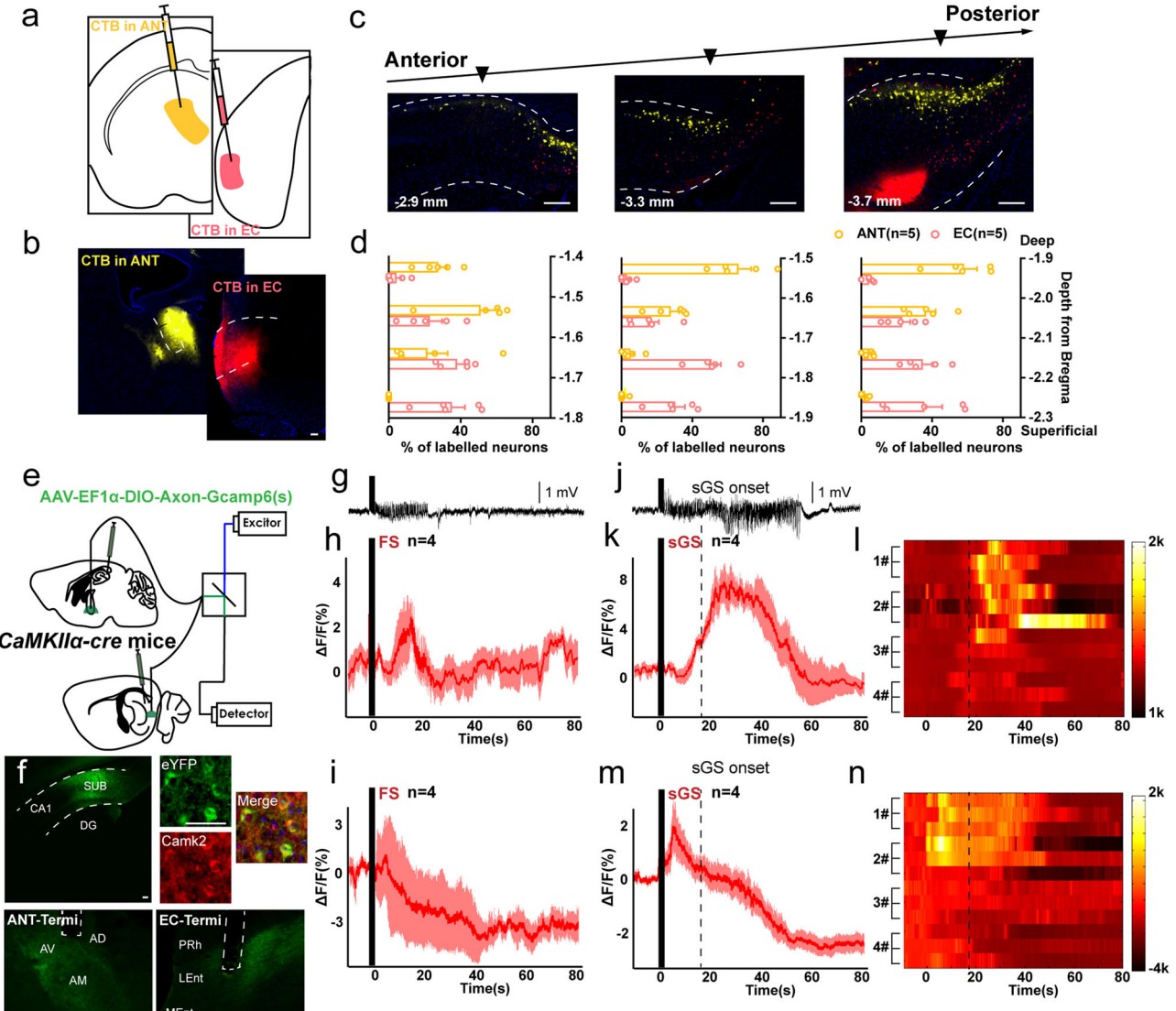

**Fig. 5 | The ANT-projecting and EC-projecting subicular neuron sub-populations have different responses to hippocampal seizures. a** Scheme of CTB injection in the anterior nucleus of thalamus (ANT) and entorhinal cortex (EC). **b** Representative images of CTB injection areas in the ANT and EC. Scale bar, 100 µm. **c** Representative images show somata locations of CTB-555 and CTB-647 in the subiculum (SUB) along the anterior-posterior axis. The number in the bottom indicated the approximate position to the bregma. Scale bar, 100 µm. **d** Quantification of CTB-555 and CTB-647 somata in the SUB, corresponding to bregma positions in (**c**). **e** Scheme of *CaMKIIα-cre* mouse injected with AAV-DIO-Ef1α-Axon-GCaMP6(s) and fiber photometry in the ANT and EC. **f** Left, representative coronal images of virus injection in the SUB and optical cannula placements in the ANT and EC in a *CaMKIIα-cre* mouse injected with AAV-DIO-Ef1α-Axon-

GCaMP6(s). Right, the overlap between GCaMP6(s) and CaMKII⁺ neurons in the SUB. Scale bar, 25 µm for the upper right images and 100 µm for the rest. **g** Typical hippocampal EEG of a mouse with focal seizure (FS), corresponding to **h** and **i**, the black bar indicated kindling stimulation period, also applied to **h–j**, **k**, **m**. **h, i** Average calcium signals (shown in $\Delta F/F$) in the ANT and EC during FSs. **j** Typical hippocampal EEG of a mouse with secondary generalized seizures (sGS), corresponding to **k** and **m**. **k** Average calcium signals (shown in $\Delta F/F$) in the ANT during sGSs. 3 trials for each mouse. **l** Heatmaps corresponding to the average calcium signals in the ANT during sGSs. **m** Average calcium signals (shown in $\Delta F/F$) in the EC during sGSs. 3 trials for each mouse. **n** Heatmaps corresponding to the average calcium signals in the EC during sGSs. The number of mice used in each group is indicated in the figure. Data are presented as means ± SEM.

terminals affects hippocampal seizures. During SUB-ANT photo-activation in sGS expression, we find that the latency to GS (initial components of GS) was barely affected, while GSD (later components of GS) was significantly extended, which indicates SUB-ANT circuit facilitating seizures through modulation of later components of GS by a longer termination. Such phenomena could be due to the reverberating loops between the thalamus and the subiculum, which need further elucidation. Interestingly, although the ANT also receives dense input from the MMB of the hypothalamus[49], our findings reveal that inhibition of the SUB-MMB circuit has no effect on the development of hippocampal seizures. Epileptic signals in the hippocampus directly pass from the subiculum to the ANT, rather than via an indirect

SUB-MMB-ANT pathway, which further highlight the ANT as an 'amplifier' in hippocampal-related epilepsy.

Another important and surprising finding of our study is that selective inactivation of the subicular neurons projecting to the EC terminal augments seizure severity in both kindling and KA models. As shown by axonal calcium signaling, the neural activity of the SUB-EC circuit was mainly inhibited during hippocampal seizures and failed to return to a basal state even long after seizure termination, indicating suppression from the upstream input to the EC-projecting subicular neurons. Correspondingly, somatostatin mRNA was reported to be rapidly and transiently expressed in the terminals of the outer molecular layer of the subiculum after KA-induced SE[44], which may account

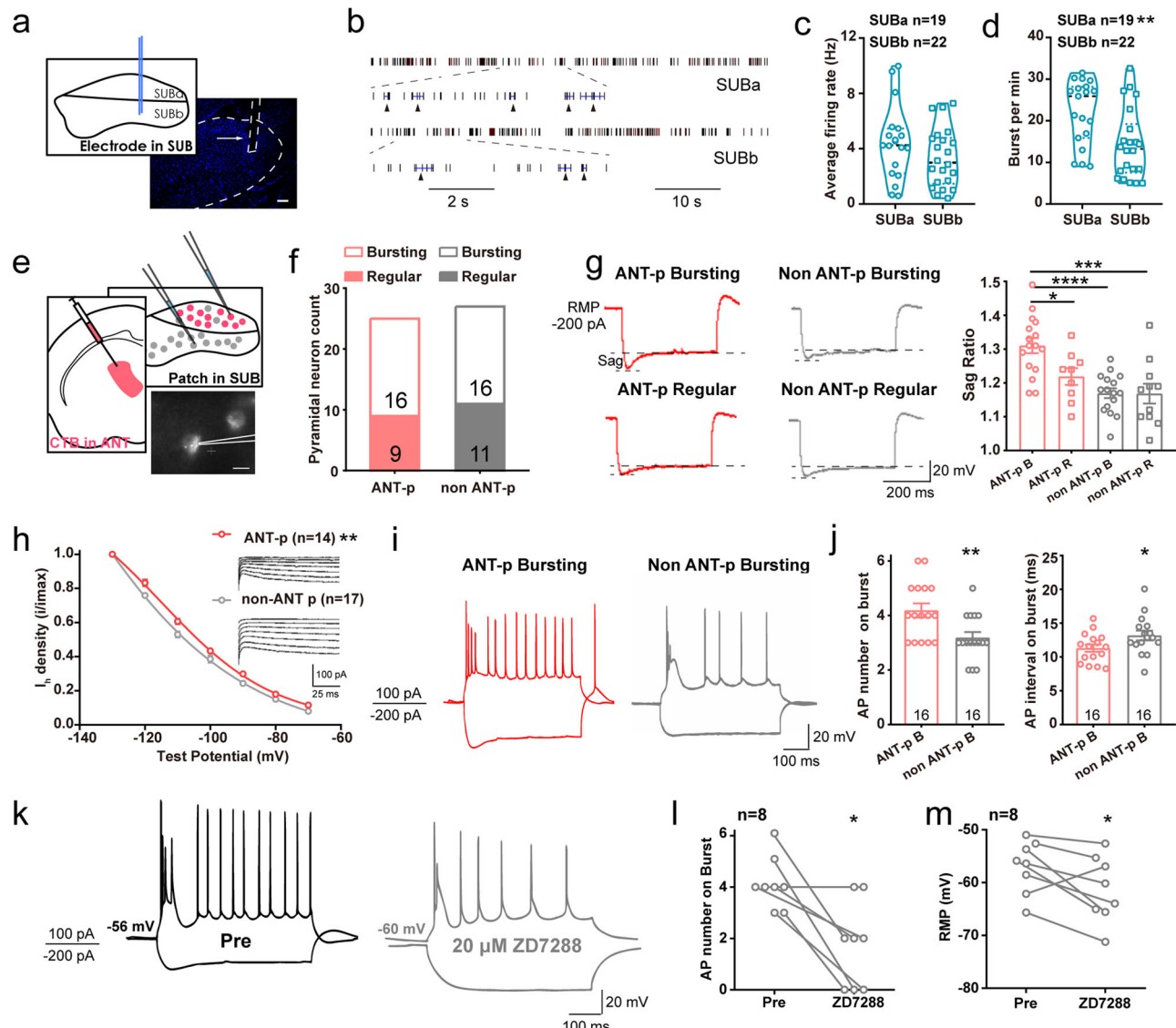

**Fig. 6 | ANT-projecting subicular neurons show stronger bursting intensity intrinsically facilitated by HCN channel. a** Scheme and representative image of in vivo recording in distinct layers of the subiculum (SUB). Scale bar, 100 μm. **b** Typical firing rate of bursting units in the SUBa and SUBb. Each subjacent triangle indicates a defined bursting. Graphs of average firing rate (**c**) and burst per minute (**d**) of bursting units in the SUBa and SUBb in the basal state. Unpaired *t*-test, **P < 0.01. **e** Scheme of in vitro electrophysiological patch in the somata of ANT-projecting and non-ANT-projecting subicular neurons and one representative ANT-projecting subicular neuron. Scale bar, 10 μm. **f** Statistics of number of ANT-projecting and non-ANT-projecting burst and regular firing neurons. **g** Left, representative voltage traces and current from patched neurons. Right, graph of the sag ratio between ANT-projecting and non-ANT-projecting bursting and regular firing neurons, *n* = 16, 9, 16, 11, respectively. One-way ANOVA with *post hoc* Tukey's

test, *P < 0.05, ***P < 0.001, ****P < 0.0001. **h** $I_h$ voltage-dependent activation curves between ANT-projecting and non-ANT-projecting neurons. Both curves fitted with Boltzmann function. Two-way repeated measures ANOVA, **P < 0.01. **i** Representative action potentials of ANT-projecting and non-ANT-projecting subicular bursting neurons evoked by −200 and 100 pA injection current. **j** Graphs of action potential number and frequency on burst between ANT-projecting and non-ANT-projecting subicular bursting neurons. Mann–Whitney test, *P < 0.05, **P < 0.01. **k** Representative action potentials in one subicular bursting neuron before and after incubation of 20 μM HCN channel blocker, ZD7288. Graph of action potential number on burst (**l**) and rest membrane potential (**m**) before and after ZD7288 buffering. Wilcoxon matched-pairs signed rank test for AP number on burst, *P < 0.05. Paired t test for RMP, *P < 0.05. The number of mice used in each group is indicated in the figure. Data are presented as means ± SEM.

for such afferent suppression. Moreover, it remains elusive why inhibiting the SUB-EC circuit promotes hippocampal seizures. The imbalance of excitation and inhibition in the EC is highly involved in ictogenesis and epileptogenesis[4]. It is suggested that optogenetic activation of both pyramidal neurons and GABAergic interneurons induced a hypersynchronous seizure onset pattern in hippocampus-EC slices[50,51]. Our previous study also demonstrated that low-frequency optogenetic activation of entorhinal pyramidal neurons retarded hippocampal seizure development[52], while inhibition of the subicular terminal in the EC might break this anti-seizure effect. In addition, both pyramidal neurons and fast spiking interneurons in the deep layer of

the EC are largely innervated by the hippocampal output[53,54], whether the subiculum-EC circuit recruits a feedback inhibition motif in epileptic seizures needs further exploration.

Recent advances have validated that subicular pyramidal neurons are anatomically and behaviorally distinct along both proximal-distal and deep-superficial axes[21,23]. Our results in CTB tracing and electrophysiological studies strongly support this idea. The ANT-projecting and EC-projecting subicular neurons are well defined in terms of their locations, and their neural activities in responding to seizures are also different. More importantly, the deep located ANT-projecting neurons showed a larger somatic $I_h$ and bursting intensity. Although we did not

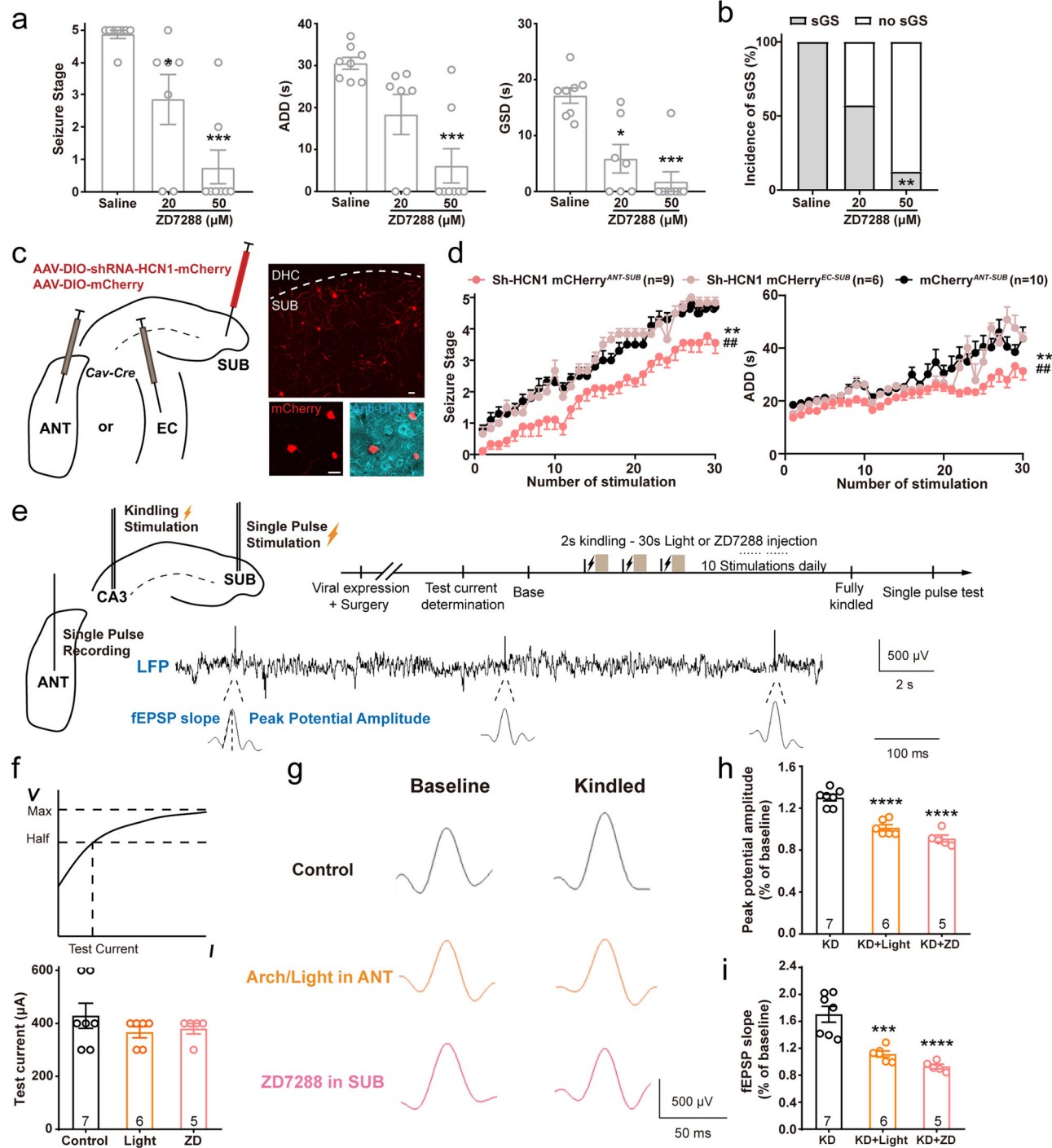

**Fig. 7 | HCN channel-based synaptic plasticity in the SUB-ANT circuit contributes to hippocampal seizures. a** Effects of ZD7288 (intra-subicular injection, 500 nL) on the seizure stage, after-discharges durations (ADD), and generalized seizure durations (GSD) of sGS expression. $N$ = 8, 7, and 9 mice, respectively. Kruskal–Wallis with *post hoc* Dunn's test, *$P$ < 0.05, ***$P$ < 0.001 compared to Saline. **b** Effects of ZD7288 on the incidence of sGS. $N$ = 8, 7, and 9 mice, respectively. Fisher's exact test, **$P$ < 0.01. **c** Left, scheme of experiments for viral expression in the ANT, EC and subiculum for conditional knockdown of HCN expression in ANT- and EC-projecting subicular neurons of wild-type mice. Right top, representative coronal image of mCherry florescence in the subiculum from a Sh-HCN1-mCherry $^{ANT-SUB}$ mouse. Right bottom, mCherry co-labeled with anti-HCN1, indicating the efficacy of knockdown. Scale bar, 25 μm. **d** Effects of HCN1 knockdown in a specific circuit on the development of seizure stage and ADD in hippocampal kindling model. Two-way repeated measures ANOVA *post hoc* Scheffe's test, **$P$ < 0.01, ShRNA-HCN1 $^{ANT-SUB}$ compared to mCherry; ##$P$ < 0.01, ShRNA-HCN1 $^{ANT-SUB}$ compared to ShRNA-HCN1 $^{EC-SUB}$. **e** Scheme of experiments for field excitatory postsynaptic potential (fEPSP) with single pulse test in the SUB-ANT circuit, and typical LFP recorded in the ANT during the test. **f** No differences were observed in test currents of the CaMKIIα-eGFP $^{SUB-ANT}$-light (Control), CaMKIIα-Arch $^{SUB-ANT}$-light (Arch), and CaMKIIα-eGFP $^{SUB ZD7288}$ (50 μM, 500 nL, ZD) groups. **g** Representative images of the single pulse response in the ANT in baseline and kindled state of three groups. Graphs of peak spike amplitude (**h**) and fEPSP slope (**i**) change in base and kindled states of three groups. One-way ANOVA with *post hoc* Dunnett's test, ***$P$ < 0.001, ****$P$ < 0.0001 compared with control group. The number of mice used in each group is indicated in the figure. Data are presented as means ± SEM.

observe high variation of burst and regular firing patterns between ANT-projecting deep located neurons and non-ANT-projecting superficially located neurons as in the previous study[55], such stronger bursting with augmented $I_h$ may make it easier for ANT-projecting subicular neurons to transmit epileptic signals. Indeed, how $I_h$ is involved in epilepsy is complicated due to its dual effects on membrane potential and input resistance. During different timings and affected areas of epilepsy patients and animal models, both increased and decreased $I_h$ were reported, providing a complex picture of whether it is a pro- or anti-epileptic factor. For example, in CA1 neurons, upregulation of HCN mRNA and $I_h$ were observed in experimental seizures[56]. Direct usage of selective HCN channel blockers in the cortex, thalamus, or systematically could inhibit seizures[57–59]. In contrast, in pyramidal neurons of the EC, impaired dendritic $I_h$ was found in rodent slices after KA-induced seizures, and global deletion of HCN1 resulted in higher propensity of epileptic behaviors[60,61]. These contradictory findings might be relevant to distinct characters of HCN channel isoforms, their distributions across brain regions, and different seizure types related, suggesting the importance of insights into region- and circuit-specific functions of HCN channels in epilepsy. Our results indicated that in the subiculum, especially in ANT-projecting pyramidal neurons, the HCN channel contributes to hippocampal seizures via enhanced bursting intensity and synaptic plasticity. It was also observed in several previous studies that strengthened HCN channel function facilitates bursting activity[62,63], leading to a possible contribution to subicular neural hyper-excitability. Thus, $I_h$ in the subiculum might act as a 'promoter' of hippocampal seizures, and interventions targeting the HCN channel in a specific SUB-ANT circuit could help future epilepsy therapy. In addition, T-type $Ca^{2+}$ channel in the subiculum is also an important factor contributing to pyramidal neuron bursting[39], pharmacological blockage of T-type $Ca^{2+}$ channel significantly inhibited bursting intensity of subicular pyramidal neurons and hippocampal seizures, which is in agreement with the previous finding[64], further confirming that bursting firing in subicular pyramidal neurons is important in generalization of seizures. Intriguingly, our finding is consistent with previous studies that bursting in subicular neurons may occur at more depolarizing membrane potentials (Fig. S12)[65,66]. This seems to be different from "de-inactivation" theory of thalamic T-type $Ca^{2+}$ current[67]. Classically, bursting firing is usually induced shortly after delivery of a hyperpolarizing pulse which let the T-type $Ca^{2+}$ channel recover from inactivation (so-called "de-inactivation"). One possible explanation could be the "window current" phenomenon of T-type $Ca^{2+}$ channel. There might be a small but obvious window component of calcium currents in the subiculum and only a small fraction of T-type $Ca^{2+}$ channel population activation was enough to generate a robust bursting near membrane potential[68]. Alternatively, bursting in the subiculum could also be driven primarily by non-inactivating, high-voltage-activated $Ca^{2+}$ tail currents by an action potential[65], distinguishing the bursting mechanisms in the subiculum from those of thalamic neurons.

Overall, we demonstrate that subicular pyramidal neurons are structurally and functionally heterogeneous, and have contrary roles in TLE via different downstream targets. In particular, an intrinsically enhanced $I_h$-contributed bursting intensity in ANT-projecting subicular pyramidal neurons facilitates synaptic plasticity and seizure generalization, which may be of therapeutic significance in understanding the neural circuit of the subiculum in TLE.

## Methods
### Animals
All procedures were approved by the guidelines of the Animal Advisory Committee of Zhejiang University and in complete accordance with the National Institutes of Health Guide for the Care and Use of Laboratory Animals. C57BL/6J (Wild-type) and *CaMKIIα-Cre* (Stock number 005539) mice were used and genotyped in line with the

protocols provided by the Jackson Laboratory. Mature male mice (8–16 weeks) were used in viral tracing, multi-unit recording, and behavior experiments. Mice at 4- to 6-weeks-old were used in in vitro electrophysiology. All mice were group-housed (four to six) in plastic cages prior to surgery with a 12 h light/dark cycle. The ambient temperature was kept about 23–26 °C and humidity was about 50–60%. After surgery, they were housed two to three per cage for better recovery. Behavior tests were conducted during the light cycle.

### Viral constructs
For calcium photometry, AAV2/9-CaMKIIα-GCaMP6(s) (viral titers: 5.6*10^12 particles/mL) and AAV2/9-EF1α-DIO-Axon-GCaMP6(s) (viral titers: 5.6*10^12 particles/mL) were injected into the subiculum of wild-type and *CaMKIIα-Cre* mice, respectively. For selective optogenetic activation or hyperpolarization of pyramidal neurons and their projection terminals, AAV2/8-CaMKIIα-hChR2-eYFP (viral titers: 1.7*10^13 particles/mL) or AAV2/9-CaMKIIα-ArchT-eGFP (viral titers: 1.7*10^13 particles/mL) were injected into the subiculum or ANT of wild-type mice. For the control group, AAV2/9-CaMKIIα-eGFP (viral titers: 1.7*10^13 particles/mL) was injected. For chemogenetic inhibition of pyramidal neuron projecting terminals, AAV2/9-CaMKIIα-hM4Di-mCherry (viral titers: 1.7*10^13 particles/mL) was injected into the subiculum of the wild-type mice. For the control group, AAV2/9-CaMKIIα-mCherry (viral titers: 1.7*10^13 particles/mL) was injected. To selectively knock down HCN1 expression in the subiculum, we used the following sequence: CCTCCAATCAACTATCCTCAA[69]. Cav2-*cre* (viral titers: 3.0*10^12 particles/mL) was injected into the ANT or EC of wild-type mice, then AAV2/9-EF1α-DIO-miR30shRNA (Hcn1)-mCherry (viral titers: 7.7*10^12 particles/mL) was injected into the subiculum. Viruses were purchased from OBio Technology (China), except for AAV2/9-EF1α-DIO-Axon-GCaMP6(s) and Cav2-*cre* from Taitool Bioscience (China). For retrograde tracing, cholera toxin subunit B conjugated to Alexa-555 or Alexa-647 (CTB-555 and CTB-647, Thermo Fisher, USA) diluted in PBS solution at a concentration of 1% wt/vol was used. 4–7 days after injections, mice were perfused for histology.

### Stereotactic injections and surgeries
For virus injections, mice were anesthetized with sodium pentobarbital (60 mg/kg, i.p.) and mounted in a stereotaxic apparatus (RWD Life Science, China). Injections were targeted into the SUB (antero-posterior (AP) −3.4 mm; lateral (L) −2.0 mm; ventral (V) −1.7/−1.8 mm), ANT (AP −0.6 mm; L −0.7 mm; V −3.5 mm), EC (AP −4.8 mm; L −4.0 mm; V −3.0 mm), MMB (AP −2.8 mm; L −0.2 mm; V −4.9 mm), and NAc (AP +1.4 mm; L −0.8 mm; V −5.0 mm) using a glass micropipette attached to a 1 μL syringe at 60 nL/min. For behavioral experiments, injection volumes were 200 nL for the SUB, 150 nL for the ANT and EC. For AAV anterograde tracing, injection volumes were 100 nL for the SUB. For CTB retrograde tracing, injection volumes were 120 nL for the ANT and EC, 80 nL for the MMB and NAc. For Cav2-*cre* injection, volumes were 150 nL for the ANT and EC. After each injection, the needle was left in place for 10 min before withdrawal.

For KA injection, mice were anesthetized with ~2% isoflurane. KA (0.25 μg in 0.5 μL saline, ab120100, Abcam) was injected into the dorsal CA1 (AP −2.1 mm; L −1.2 mm; V −1.6 mm) with a glass micropipette attached to a 1 μL syringe at 100 nL/min. After each injection, the needle was left in place for 10 min before withdrawal.

For implanting electrodes or optical cannulas, mice were anesthetized with ~2% isoflurane three weeks after viral delivery, and then twisted-bifilar stainless electrodes (795500, 0.127 mm diameter, A.M Systems, USA) were implanted into the right hippocampal CA3 (AP −2.9 mm; L −3.1 mm; V −3.1 mm) for both kindling stimulation and EEG monitoring. Optical cannulas (0.2 mm diameter, Inper, China) were separately lowered into the following areas: SUB (AP −3.4 mm; L −2.0 mm; V −1.7/−1.8 mm), ANT (AP −0.6 mm; L −0.7 mm; V −3.5 mm), EC (AP −4.8 mm; L −4.0 mm; V −3.0 mm), MMB (AP −2.8 mm; L

−0.2 mm; V −4.9 mm), and NAc (AP + 1.4 mm; L −0.8 mm; V −5.0 mm), allowing for light stimulation. Cannulas (0.41 mm diameter, RWD Life Science, China) were implanted into the SUB, ANT, and EC for drug delivery. Then three screws were placed over the skull to fix the dental cement, two of which were placed over the motor cortex and cerebellum to serve as the ground and reference electrodes, respectively. Viral expressions and locations of the electrodes were verified after all behavior tests. We only included mice with correct electrode, cannula emplacement, and viral expression (approximately 30% mice were excluded under these criteria).

### Calcium fiber photometry

Fiber photometry was performed 1 week after electrode and fiber implantation surgeries in mice expressing Gcamp6(s). The fiber photometry system contains a 488 nm diode laser (OBIS 488LS, Coherent, USA), a dichroic mirror (MD498, Thorlabs, USA), and coupled into an 0.23 mm, 0.37 NA optical fiber with a 10× objective lens (Olympus) and a fiber launch (Thorlabs). The power of the laser intensity between the fiber tip and the mice brain regions ranged from 0.01 to 0.03 mW to avoid bleaching. The collected GCaMP fluorescence was converted to voltage signals by a digital amplifier (C7319, Hamamatsu, Japan). The converted signals were recorded at 100 Hz for 200 s (100 s baseline and 100 s after kindling stimulation). Data were further analyzed by MATLAB (version R2017b, MathWorks, USA). The values of fluorescence change were shown as $\Delta F/F$ with the following equation: $(\Delta F/F) = (F − F_0)/F_0$, among which $F$ represented the current value of signal, $F_0$ represented the average value of baseline signals between 90 and 100 s. The data are presented as time-related peri-event plot and heatmap.

### Hippocampal kindling model

For rapid hippocampal kindling model, after 1 week of recovery, EEG of the right hippocampal CA3 was recorded by a Neuroscan system (Compumedics, Australia). Kindling stimulations were all conducted during wakefulness. The after-discharge threshold (ADT) of each mouse was determined (monophasic square-wave pulses, 20 Hz, 1 ms/pulse, 40 pulses) by a constant current stimulator (SEN-7203, SS-202J, Nihon Kohden, Japan) as in our previous study[16]. The stimulation current started at 40 μA and then increased 20 μA each time, and the minimal current that produced at least 5 s ADD was defined as ADT. Only mice with ADT less or equal than 200 μA were used later. All mice received ten suprathreshold stimulations (monophasic square-wave pulses, 400 μA, 20 Hz, 1 ms/pulse, 40 pulses, 30 min-interval per stimulation) daily from the next day on. Seizure severity was scored following the criteria of the Racine scale[70]: 1. facial movement; 2. head nodding; 3. unilateral forelimb clonus; 4. bilateral forelimb clonus and rearing; and 5. rearing and falling. Stages 1–3 are considered as FSs and stages 4–5 are sGSs. The length of ADD was defined as the duration between the moment of kindling stimulation and the end of paroxysmal discharge event in EEG. These continuous discharges showed an average amplitude >3 times versus baseline and isolated postparoxysmal spikes were not calculated in the ADD according to previous study[70]. Behavioral assessment and EEG analysis were performed by well-trained experimenters blinded to the group allocation. When mice had three successive stage 5 seizures, they were regarded as fully kindled. During this period, mice exhibited stable sGS after each kindling stimulation. To determine the effect of specific interventions on sGS, we used suprathreshold current (200 μA, 20 Hz, 1 ms/pulse, 40 pulses, 30 min-interval per stimulation) to induce sGS. The EEG power recorded was calculated and analyzed offline by a software package (Scan 4.5) in the Neuronscan System. For electrical lesion studies, we used 1 mA, 10 s, direct current stimulation in the subiculum.

### Photo stimulation

Blue (473 nm) or yellow (589 nm) light was delivered by a 200 μm diameter optical fiber (Inper) connected to the laser by a Master-8

(AMPI, Israel) commutator. The optical fiber was cut flat, and power of the laser was adjusted to 5 mW. Immediately before the mouse was placed in the chamber, the cannula cap was removed, and an optical fiber was directly inserted. In the hippocampal kindling model, optical stimulation was delivered immediately after the end of kindling stimulations. For optogenetic activation experiments, 473 nm blue light (20 Hz, 10 ms/pulse and 600 pulses) was used. For optogenetic inhibition experiments, 589 nm yellow light (continuous 30 s) was used.

### Intra-hippocampal KA model

In this model, SE is typically induced via intra-hippocampal injection of KA (as mentioned in 'Stereotactic injections and surgeries' section), which further caused spontaneous recurrent seizures in the following several months. Two months after KA and AAV-CaMKIIα-hM4Di-mCherry or AAV-CaMKIIα-mCherry injection, EEG of the hippocampal CA3 was continuously recorded in freely moving surviving mice through a Powerlab system (AD Instruments, Australia) at a sample rate of 1 kHz, which was synchronized with video monitoring 8 h/day for 3 days as baseline (Pre). During this period, each mouse was intra-ANT or EC injected with saline for 500 nL daily before recording start. A FS was defined as a sharp paroxysmal event that continued more than 10 s and had an average amplitude > 3 times versus baseline and frequency > 2 Hz. A sGS was defined as an event that continued more than 30 s and had a period of post-inhibition, accompanied by a marked tonic-clonic behavioral seizure[71]. Only mice with detectable seizures were given CNO injections (The percentage of mice with detectable seizures in the study was about 40–50%). These mice were locally injected with CNO (1.0 mM, 500 nL in the ANT or EC) daily for 3 days to test the effect of chemogenetic inhibition of ANT- or EC-projecting subicular terminals on spontaneous seizures. Mice with mCherry were also given CNO via cannula to test its effect on seizures. The concentration and volume of intra-brain CNO injections were referenced in previous studies[72,73]. Mice were then injected with saline in the next 3 days for post treatment.

### In vivo multi-unit recording and analysis

Multi-unit recording[16] was performed as following: the body temperature of mice was kept at 37 °C by a heating pad. 12 microelectrodes (761500, 0.025 mm, A.M Systems) were twisted into a bundle to form recording electrodes, which had an impedance of 1–2MΩ and were combined with an optical fiber terminal to keep stiff. Neuronal activities were sampled by the Cerebus acquisition system. Recorded units were identified by low firing rate (<10 Hz), wide spike waveform (>0.3 ms) and flat auto-correlograms. They were post positioned for their sub-regional locations in the SUBa (the deep part of the subiculum) or SUBb (the superficial part of the subiculum). In particular, when electrodes were positioned at AP −3.3 to −3.5 mm, units at depths of −1.30 to −1.65 mm were located in SUBa and units at depths of −1.65 to −1.90 mm were located in SUBb; when electrodes were positioned at AP −3.5 to −3.7 mm, units at depths of −1.70 to −1.80 mm were located in SUBa and units at depths of −1.80 to −2.00 mm were located in SUBb; the rest that were not in these locations or whose locations could not be identified were excluded. A firing pattern that had more than three spikes with less than 170 ms interval was defined as one burst. Each burst had more than 200 ms external interval. We considered units with >5 bursts per min as bursting units[74].

### Subiculum slice preparation

4 weeks old wildtype mice were injected with CTB-555 3–5 days before slice experiments. Under sodium pentobarbital (100 mg/kg, i.p.) anesthesia, they were decapitated, and the brain was quickly removed and placed in an ice-cold solution containing (in mM): 110 Choline chloride, 2.5 KCl, 1 $NaH_2PO_4$, 0.5 $CaCl_2$, 7 $MgCl_2$, 20 Glucose, 1.3 Ascorbate acid, 0.6 Na-pyruvate, 25 $NaHCO_3$ (pH 7.4, oxygenated with 95% $O_2$ and 5% $CO_2$). The 300 μm coronal or sagittal slices containing the hippocampus were cut using a vibratome (VT1000, Leica, Germany). Slices

were then placed into a chamber filled with artificial cerebrospinal fluid (ACSF) containing (in mM): 120 NaCl, 11 Dextrose, 2.5 KCl, 1.28 $MgSO_4$, 2.5 $CaCl_2$, 1 $NaH_2PO_4$, and 14.3 $NaHCO_3$ (pH 7.4, oxygenated with 95% $O_2$ and 5% $CO_2$) and incubated at 34.7 °C for 0.5 h and then maintained at 25 °C for further experiments.

## In vitro electrophysiology and analysis

For slice recordings, slices were kept at 25 °C in a recording chamber perfused with 3 mL/min ACSF, which contains the same substances as the storage ACSF. Patch pipettes were pulled from glass capillaries and at resistances of 6–9 MΩ, which contained (in mM): 35 K-gluconate, 110 KCl, 10 HEPES, 2 $MgCl_2$, 2 $Na_2$ATP, and 10 EGTA. Patch-clamp recordings were performed by an EPC10 patch-clamp amplifier (HEKA Instruments, Germany), with a low-pass filter at 3 kHz, and a sample rate of 10 kHz. The series resistance and capacitance were compensated after a stable Gigaseal. Recordings were typically performed 3 min after break-in. Data were analyzed by Clampfit software (Molecular Devices, USA).

For current–voltage ($I/V$) curves, hyperpolarizing to depolarizing current gradients (−250 to 250 pA, 50 pA each step, 500 ms) were injected. To plot the number of bursting spikes as a function of membrane potential, patch recordings were additionally performed varying the membrane potential from −55 mV to −70 mV at gradient of 5 mV. The voltage sag ratio was calculated with the following equation at −200 pA: sag ratio = $(V_{base} - V_{min})/(V_{base} - V_{steady})$, among which $V_{base}$ represented the resting membrane potential, $V_{min}$ is the hyperpolarizing current that induced minimum voltage, and $V_{steady}$ is the averaged voltage caused by the inject current. The numbers and frequency of burst AP were calculated at the 100 pA depolarizing inject current. Input resistance ($R_{in}$) was calculated from the gradient in the linear phase of current–voltage plots following responses to hyperpolarizing step current injection. The membrane time constant was fit by an exponential function of the membrane potential change in response to rectangular hyperpolarizing current injection that induced small (3–5 mV) voltage deflections.

For AP properties, depolarizing current pulses (increased 5 pA each step) were applied to measure the threshold, amplitude, and the half-wave width of the APs. We analyzed the first spike (first single spike for bursting neurons) induced by the minimum depolarizing current (Rheobase). AP amplitude was defined as the voltage from the AP threshold to the AP peak, and the AP half-width was calculated as the duration at half-maximal amplitude.

For HCN channel properties, tail currents were measured by stepping the membrane potential from an initial holding potential of −60 mV to test potentials of −130 mV to −60 mV in 10 mV increments at an interval of 200 ms. In slice pharmacology, recordings were performed 20 min after incubation of 20 µM ZD7288 (ab120102, Abcam) or 5 µM TTA-P2 (T-155, Alomone) until an apparent steady-state effect was achieved.

## Single pulse measurement

For single pulse measurement, an opto-electrode was implanted into the ANT of CaMKIIα-Arch [SUB] or CaMKIIα-eGFP [SUB] mice for EEG recording and light delivery, a cannula-electrode was implanted into the subiculum for single pulse stimulation and drug injection, and an electrode was implanted into the CA3 for kindling stimulation. After recovery for 1 week, experiments included three parts[75]. First, the test stimulation current was determined before CA3 kindling. The stimulation intensity in the subiculum that induced half of the maximum voltage in the ANT was considered as the test current. Second, all mice received ten suprathreshold stimulations (monophasic square-wave pulses, 400 µA, 20 Hz, 1 ms/pulse, 40 pulses) daily from the next day on until they were fully kindled (three successive stage 5 seizures). During this period, the first group of CaMKIIα-eGFP [SUB] mice and CaMKIIα-Arch [SUB] mice received 30 s yellow light stimulation in the ANT after each kindling, and the second group of CaMKIIα-eGFP [SUB]

mice were intra-subicular injected with ZD7288 daily before the kindling stimulation. Finally, the single pulse test was performed (monophasic square-wave pulses 100 µs/pulse, 10 s interval, 10 pulses). The results of the single pulse test were compared before and after sGS acquisition in control, optogenetic inhibition in the ANT of CaMKIIα-Arch [SUB] mice, and ZD7288 (50 µM, 500 nL) injection in the subiculum. Peak potential amplitude was defined as the voltage from the field EPSP (fEPSP) threshold to the fEPSP peak. fEPSP slope was defined as the ratio from peak potential amplitude versus the time from the fEPSP threshold to the fEPSP peak.

## Histology and quantification

Mice were deeply anesthetized with pentobarbital and transcranially perfused with 0.9% saline and followed by 4% paraformaldehyde in 0.1 M phosphate buffer. Notably, in c-Fos staining experiments, animals were sacrificed 1.5 h after seizures. Brains were removed and stored in the same fixative overnight at 4 °C and then dehydrated in 30% sucrose. Coronal or sagittal sections were cut on a freezing microtome (Themo Scientific, USA) at 40 µm. The primary antibodies used were as follows: rabbit monoclonal anti-CaMKII (1:800; ab134041, Abcam), guinea pig polyclonal anti-c-Fos (1:2000; 226004, Synaptic Systems), rabbit monoclonal anti-NeuN (1:800; MABN140, Millpore) and mouse monoclonal anti-HCN1 (1:500; MAB6651, Abnova). The primary antibodies were incubated overnight at 4 °C and rinsed with PBS. Then Alexa-488, Alexa-594, or Alexa-647 conjugated secondary antibodies were used at 1:800 for 2 h at room temperature. The secondary antibodies used were as follows: Alexa Fluor® 488 or 594 AffiniPure Donkey Anti-Rabbit IgG (H + L) (Jackson ImmunoResearch, 711-585-152 or 711-545-152), goat Anti-Guinea pig IgG H&L (Alexa Fluor® 647) (ab150187, Abcam), goat Anti-Mouse IgG H&L (Alexa Fluor® 647) (ab150115, Abcam). Sections were then washed and mounted on slides with media containing DAPI (Vectashield Mounting Media, Vector Labs). Images were captured by a confocal microscopy (SP8, Leica).

Image analyses and quantification were performed using ImageJ (version 1.52a) software. For c-Fos[+] and NeuN[+] cell quantification, we counted the number of cells that exhibited c-Fos[+] or NeuN[+] immunoactivity within the corresponding nucleus of three representative coronal slices (anterior, intermediate, and posterior), and then calculated the mean value for each mouse. The number of mice used in each experiment was indicated in figure legends.

## Statistics and reproducibility

Data are presented as means ± SEM. All experiments were repeated at least two times independently with similar results. Data analysis was performed blind to the experimenters using SPSS (version 17.0, IBM) and Prism 8 (Graphpad Software). Data with Gaussian distributions were analyzed by one-way or two-way ANOVA followed by post hoc Dunnett's, Tukey's or Scheffe's test for multiple comparisons, unpaired t-test, or paired t-test for statistical significance when appropriate. Non-normally distributed data were analyzed by Friedman with post hoc Dunn's test for multiple comparisons, Kruskal–Wallis test or Wilcoxon test. Incidence data was analyzed by Fisher's exact test. Significance is reported in the figure legends. For all analyses, the tests were two-tailed and the $P$-value $<0.05$ was considered statistically significant. Detailed statistic parameters are provided in the Supplementary Table S2 with the paper.

## Reporting summary

Further information on research design is available in the Nature Research Reporting Summary linked to this article.

## Data availability

The authors declare that all data supporting the findings of this study are available within the paper and its supplementary information files "Source Data". The source data are provided as a Source Data file with the paper. Source data are provided with this paper.

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

## Acknowledgements
This project was supported by grants from the National Natural Science Foundation of China (Grant No. 82022071) (Y.W.), (Grant No. 81630098, Grant No. 81821091) (Z.C.), (Grant No. 81971208) (Y.D.) and Natural Science Foundation of Zhejiang Province (Grant No. LD22H310003) (Z.C.).

## Author contributions
Y.W. and Z.C. initiated and coordinated the project; Y.W., F.F., and C.X. designed the experiments; F.F., X.W., C.X., J.S., and Y.G. conducted the experiments; F.F., X.W., and C.X. analyzed the data; H.C., N.L. Y.R. Y.D., and S.W. contributed to data discussion; F.F., Y.W., and Z.C. wrote the manuscript.

## Competing interests
The authors declare no competing interests.
