## [Peer Review File · Nature Communications]

REVIEWER COMMENTS

Reviewer #1 (Remarks to the Author):

The ms “Deconstructing the Diversity of Subicular Circuits in Seizure Generalization of Temporal Lobe Epilepsy” by Fan Fei and collaborators is an extension of their previous study on how the subiculum is a gatekeeper of temporal lobe seizures generalization. Although the role of subiculum in the initiation and propagation of hippocampal seizures has been shown before, the exact cellular mechanisms, as well as downstream propagation of seizures remained elusive. Here they show that the subiculum itself may generate seizures and that subicular heterogeneity should be taken into consideration and uncover the role of subicular neural circuits with different projections. Indeed, their results clearly show that subicular neural circuits differently modulate hippocampal seizures, in particular, anterior thalamus nuclei-projecting neurons bidirectionally modulates seizures through HCN channel associated mechanisms, while Entorhinal Cortex-projecting circuits attenuate seizure strength. Two in vivo models are studied: a rapid hippocampal kindling and hippocampal kainate mouse. The circuits at play in epileptic discharges are labeled and then modulated by opto and chemogenetics.

The data are new and convincing. They shed a new light on the role of the subiculum in temporal lobe epilepsy. The amount of work, of data and the range of techniques used in the paper is impressive. The logic of the paper is very clear and well maintained through the text. Figures show results which often involve complex experimental design in a comprehensive manner. However, I find that text could benefit from some additional information regarding used methods, as well as some additional statistics. I support the paper publication with limited modifications.

The term “deconstructing” in the title is a bit overstated.

Figure 1.

b&c. I did not fully understand the scales of the heat map. Is the duration similar to the left traces? Was the average Ca signal normalized? May the authors distinguish trials for each animal? Was the surface activated during FS vs GS different? Where were the first signals recorded within the subiculum in FS vs GS?

g. Was the same activation pattern observed for FS?

Figure S1. May the authors provide images of the subicular damages?

Seizure stage / Racine scale. As far as I have understood it, the Racine scale is used to quantify seizure stages. How can Racine scales values (in Supplementary Fig 1, Fig 1 ...) be decimals and not 1-2-3-4 or 5?

Figure S3. The ability to elicit seizures by only stimulating subicular pyramidal cells is an important result. How did superficial vs deep layers stimulation affect seizure generation? Where some spontaneous seizures recorded after the kindling procedure?

Figure 2. What is the mechanism of the facilitation of anterior thalamus activity on subicular discharges (apart from Ih)? Is it associated with an hippocampal effect? Is it related to a direct effect on pyramidal cells properties? Is it related to reverberating loops between the thalamus and the subiculum? The later hypothesis seems reinforced by the appearance of seizure restart when the thalamus is activated (Fig2a) or by the appearance of a faster termination when the thalamus is inhibited. A way to address this question would be to quantify the activity at the beginning of the seizure in each condition and evaluate if thalamus activity modulation rather affects initial vs later components of the seizure.

May the authors explain in the legend the meaning of AD, AV, AN?? What does that mean?

b: any explanation for the drop happening in the 10 stim both for ADD and sz stage?

Figure 3. The anatomical limits of the entorhinal cortex vs subiculum are not clear (3a). May the authors provide an anatomical image in addition the fluorescent one?

“We still observed some level of cellular damage in subicular pyramidal neurons” (P9): please indicate percentage of cell loss and how it was assessed.

Figure 4. Since the EC is very close to the subiculum, how can the author validate that CNO did not affect directly subicular neurons. Further, did the authors test the effects of CNO, known to be epileptogenic, in WT KA mice?

In addition to time-frequency displays, may the authors show examples of seizures in control vs CNO condition?

Figure 5.

Page 10: “We observed that the somatic locations of the ANT-projecting neurons and the EC-projecting neurons in the subiculum were totally segregated. Specifically, ANT-projecting neurons were almost completely deep located while EC-projecting neurons were mostly distributed in the superficial layer. Such topological distribution was consistent across the anterior-posterior axis (Figure 5c and d)”. It’s unclear from the text whether this was shown before or it’s an original finding of this study.

Figure 6.

h. May the authors display a picture of the fluorescence pattern after CTB injection in the anterior thalamus (rather than a cartoon).

i. The color code is disturbing since red here stands for bursting while it is used for cells connected to the ANT in the rest of the figure. Please change.

Figure 7e. The 2 last FP examples are not aligned with the trace.

Does the HCN activator antiepileptic drug Lamotrigine worsen seizure spread?

Methods.

Page 21: "We only included mice with correct electrode, cannula emplacement, and viral expression". It would be nice to mention the percentage of mice which were excluded from analysis based on these criteria, in order to show the feasibility of these experiments.

Page 21: "Only mice with detectable seizures were given CNO injections". Please indicate the percentage of mice with detectable seizures

Page 23: "Recorded putative pyramidal neurons were post positioned for their sub-regional locations in the SUBa (the deep part of the subiculum) or SUBb (the superficial part of the subiculum)". Please clarify in the methods how neuronal positioning was verified.

Page 24 - 3.3 mM CaCl₂. Why calcium concentration in ACSF was so high and what was the composition of ACSF in which recordings were performed? If the same as storage ACSF, it should be specified.

Minor.

Page 3. "which is the common origin of epileptic seizures during TLE8". I am not sure that the reference (on comorbidities) is the good one.

Page 4: "An intrinsic enhanced I_h-dependent burst intensity, underlying the synaptic plasticity of ANT-projecting subicular pyramidal neurons, facilitates the generalization of hippocampal seizures, which may be of therapeutic significance due to an improved understanding of the subicular neural circuit in TLE." Facilities probably should be facilitates

Figure 1b. Kindling stimulatoin should be corrected.

Fig 5g & j : please add that this relates to FS

Page 14: "The subiculum is thought to be essential in the initiation and propagation of seizures in epilepsy, but how long-projecting subicular pyramidal neurons and related circuits were recruited was unknownrarely revealed before". "unknownrarely revealed before" should be rephrased.

Reviewer #2 (Remarks to the Author):

The current study explores the role of principal cells from the subiculum in temporal lobe epilepsy (TLE). Using two different models of TLE (kindling and kainate) combined with optogenetics, chemogenetics, calcium imaging, tracing and electrophysiology, the authors show that the stimulation of pyramidal neurons projecting to the anterior nucleus of the thalamus (ANT) promote epileptic seizures while their inhibition reduced the severity of the seizures. By contrast, the selective silencing of pyramidal cells projecting to the entorhinal cortex (EC) facilitated seizures. ANT projecting neurons express a hyperpolarization-activated current (I_h) and are capable of firing action potentials in bursts. Knocking down the expression of I_h alleviates the development of seizure.

The topic is important, the results are interesting and most of them are well supported by an impressive range of state-of-the-art techniques. However, I have some concerns.

Specific comments

- Why did not the authors test the effect of an activation of EC, NAc and MMB projecting neurons (as they did for ANT projecting neurons)?
- Fig 5k What does the Ca increase in EC terminal visible before GS onset correspond to?
- Optogenetic activation and inhibition of subiculum principal cells: the authors should document the response to blue light at cellular level, ideally by patching neurons. They should provide evidence that the stimulation of Arch in the projecting regions of subicular neurons inhibits neurotransmitter release and does not revert the effect of presynaptic GABA receptors due to the accumulation of Cl⁻ in the intracellular space (see PMID 22729174).

- Slice electrophysiology: The burst firing examples in fig 6l and 6n seem to be superimposed on a low-threshold calcium spike produced by a T-type calcium current. T-channels (CaV3) are important for burst firing in the subiculum (e.g., PMID: 28744923; 28326015). In addition, it was shown that the compound used in the study for blocking Ih (ZD7288) is not selective as it also blocks T-type calcium current (PMID: 18534748). The authors should test for the possible contribution of CaV3 channels to the bursting behavior and to the induction of epileptic seizures.

- In favor of a contribution of Ih, the authors show that silencing the expression of HCN1 in ANT projecting neurons. However, Ih contributes to the resting membrane potential of several neurons (e.g., PMID 9644226; 18216232) and silencing it could result in a hyperpolarization of pyramidal neurons which would also inhibit the activation of ANT. To validate their hypothesis, the authors should show that silencing HCN is sufficient to eliminate burst firing.

Minor comments

- Fig. 1b: typo

- p. 6, first paragraph: "the pro-seizure effect of optogenetic activation of subicular pyramidal neurons was reflected by the increasing number of stimulations needed..." I guess the authors mean the opposite, as shown in fig 1jk.

Reviewer #3 (Remarks to the Author):

Fei et al. investigated the role of excitatory subicular neurons in the generation of epileptiform activities. By primarily using the kindling model of epilepsy in mice, the authors measured calcium signaling in the subiculum during seizures. They manipulated the excitability of the subiculum or its downstream terminals using optogenetics or chemogenetic tools targeted to excitatory neurons. The major strength of the study is the use of a variety of methods trying to answer the same question. The data show:

1. Increased calcium signalling in excitatory subicular neurons at the onset of electrographic seizures.
2. Behavioural seizures are more severe when subicular pyramidal neurons are activated optogenetically and less severe when hyperpolarized.
3. Optogenetic activation of excitatory, subicular terminals in the anterior thalamus (SUB-ANT) worsens seizure severity while hyperpolarization of SUB-ANT lowers seizure severity.
4. SUB-ANT projections originate in laterodorsal subiculum and subiculo-entorhinal (SUB-EC) projections originate in posteroventral subiculum. Calcium response during seizures reaches its maximum earlier in SUB-EC neurons than in SUB-ANT neurons although the signal in SUB-EC neurons is weaker.

Merits

In studying subicular outputs, the authors have identified an important pathway in epileptogenesis and have performed extensive experiments to investigate the underlying mechanisms. The experiments are generally well-designed to address the hypotheses outlined in the text. Although many unclear aspects of results and their description are present in the manuscript, the main conclusion will likely stay unchanged: subicular projections to anterior thalamus boost seizure and subicular projections to entorhinal cortex dampen seizures.

Major critique.

1. The title is misleading. According to International League Against Epilepsy, epilepsy is characterized by unprovoked seizures. The main method used here to trigger seizures is electrical kindling, which is by definition stimulation triggering paroxysmal response and it is not spontaneous.
2. Electrophysiological part of the study is either not properly done or not properly described. No reliable conclusions can be made from this part of the study (details are below).
3. Separation of focal and generalized seizures is misleading. First, FS are described as stage1-3 on Racine scale, then as paroxysmal event lasting 10-30 sec. According to accepted definition, focal seizures occupy just one area of the brain and generalized seizures bilateral and occupy multiple brain areas. In the study, there was no even attempt to obtain data enabling to characterize seizures as either focal or generalized. There is also a terminology like hippocampal seizures that is used in conjunction with generalized seizures. Hippocampal seizures cannot be generalized. Generalized seizures can be primarily hippocampal sans secondary generalised.
4. Controls are often absent.
5. It was often impossible to know how many animals, slices etc were used for the analysis.
6. Method section does not give sufficient details, therefore often, it was impossible to evaluate accuracy of provided results.

Specific critique.

1. As presented, the results are convincing in demonstrating the link between subiculo-thalamic output and seizure severity. However, the lack of detail in the methods section casts doubt on the validity of the results. For example:
 - a. It is unclear how seizure scoring was performing. The authors state that they used the Racine scale but there is no detail as to the criteria used to detect behaviour such as facial movements, myoclonic jerks etc. Was the scoring performed automatically? How many facial movements constituted stage 1? If done manually, was the testing blinded and/or randomized?
 - b. How were after discharge durations calculated? What constituted a single-event? For example, in Figure 1L:eYFP-GS, do we see one long ADD or two separate ADDs according to the authors' criteria? In Figure 4c, what is shown as a focal seizures is indistinguishable from theta activity which we expect in hippocampal recordings.

c. Was kindling performed only during wakefulness? Sleep is a strong factor influencing the expression of seizures and it constitutes a major confound.

d. Which AAV serotype was used for tracing and optogenetic experiments? Most AAVs have retrograde properties. EC is reciprocally connected with the subiculum such that AAV injections in the subiculum can be taken up by EC terminals and transported retrogradely to the EC (i.e. we this somewhat in FigS4 with puncta in EC and RSG indicating cell bodies). Thus optogenetic stimulation in EC would modulate subicular neurons, possibly explaining the contrary results obtained from EC stimulation.

e. KA model of epileptogenesis is not described.

f. How was cFos counted? How many animals? Slices? Slices per animal? If only one slice, how this slice was chosen?

g. c-Fos signalling degrades fast. What was the time between seizure and euthanasia for C-fos experiments?

h. How was power calculated in Figure2g?

i. Chemogenetic experiments seems do not have any controls. Was there CNO injection without virus tested? Was there saline injection in animals with viruses expressed in target structures?

Without the above details, the validity of the data is impossible to judge.

2. Please include low-magnification histological sections both in the main body and supplemental figures along with the current pictures. It is hard to determine where injections were made without seeing anatomical landmarks.

3. The separation of the ictal broadband into alpha, beta, theta, delta and gamma is not appropriate. These rhythms are activity and state-dependent and are likely not all present within the seizure event. The result may reflect change in the overall power of the signal rather than a change in the power of individual frequency bands.

4. Page 6- "We found that activation of the subicular pyramidal neurons prolonged GS durations (GSDs), leaving seizure stages and ADDs unaffected (Figure 1n-p). In contrast, inactivation of subicular pyramidal neurons reduced GSDs (Figure 1q-s)." Did you do any controls here?

5. Fig. S4. I do not certainly understand what we see in B. If this is anterograde tracing, we should see only axons in target structures. The signal in most of the cases is so strong, that it looks like the whole structure is composed of exclusively subicular axons, which cannot be true.

6. Fig. S5 and all related text. Was there any attempt for quantification of data? Fig.S4 suggests that there are very strong projections from Subiculum to LD thalamic nucleus. Despite this fact, nothing in the study addresses a role of LD and here, it would be important to see at least cFos expression in LD?

7. The author state that they recorded 'single unit activities'. This is not true. At maximum, they recorded multiunit activities with post processing aiming to obtain information on single unit firing.

Definitely, in the type of experiments done here, it is impossible to obtain information on single unit firing derived from multiunit recordings. As described by 'fathers' of spike sorting (Harris KD, Henze DA, Csicsvari J, Hirase H, Buzsaki G (2000) J Neurophysiol 84 (1):401-414) the major errors of sorting come (a) in bursting neurons, because the amplitude and shape of every spike produced by the same cell is different and (b) the errors of detection are high when neurons fire synchronously, like in your study, during paroxysmal discharges. Thus, Spike sorting procedures cannot give any reliable information when applied to bursting neurons that tend to fire synchronously.

8. Neuronal firing activities were recorded with 12 microwire bundles combined with optical fiber. If one assumes the use of a glue or epoxy to keep all this together, the object inserted into the brain is in the order of 1 mm, which typically produces major damage to the brain including cutting all passing fibers.

9. Classification of neurons with increased/decreased activities with a criterion 20% is not acceptable. Statistical tests are regularly done in other labs to evaluate either increase or decrease in firing.

10. The use of in vitro electrophysiological recordings in this study goes below any critique. The bath solution here contained 2.5 KCl and 3.3 CaCl₂, which is more less typically used in vitro. However, during seizures the extracellular K goes to 12-16 mM and extracellular Ca to 0.6 mM. Needless to say, that Ca is major ion mediating bursting in pyramidal cells and K one of the major ions contributing to Ih. In addition, the membrane potential is not indicated making impossible to judge whether observed changes were due to specific action of ZD7288 or associated this it changes in the membrane potential.

11. Page 11: "we subsequently defined the location of each neuron in the deep or superficial with post-verification". Please include these data.

12. Page 12. "Since in vivo recording lacked the precision of SUB-ANT or SUB-EC circuits, ...". This is your fault. You could easily use antidromic stimulation in order to identify projecting neurons.

13. Page 12. "...reflecting enhanced function of the somatic hyperpolarization-activated cyclic nucleotide-gated cation (HCN) channel." What means enhanced function of a channel? The channel can be either open or closed.

14. Page 12, Fig. 6 I, n. "...ANT-projecting bursting neurons showed a higher burst firing intensity..." If the resting membrane potential is similar in shown traces, there is obviously different extent or type of high-threshold Ca channels that mediate difference in burst properties.

15. Experiments with ZD7288 injection. Was there any control done?

16. Page 14. "Both blockage of subicular HCN channels and optogenetic inhibition of the SUB-ANT circuit suppressed..." How optogenetic inhibition was achieved here?

17. Figure 7e-i cannot not be published in its present form. The authors state that they have measured field potentials in ANT elicited by subicular stimulation. However, it appears that the tentative evoked potentials are traces chosen arbitrarily in the period after the stimulation artefact (Figure 7e). Further, the field potentials have a half-width of about 30 microseconds, which is biologically meaningless. Evoked potentials elicited by SUB stimulation should not be referred to as spikes to avoid confusion with single-unit (spike) terminology.

18. Page 24. "Wildtype mice were injected with CTB-555..." What is CTB-555?

19. In the discussion and figure, the phrasing “hyperexcitatory neurons” suggests the existence of a unique population of hyperexcitatory neurons in the subiculum. Subicular calcium signaling indeed increased during seizures but it is not demonstrated that these neurons are hyperexcitatory in their normal firing.

20. The authors suggest that subicular neurons gate the generalization of hippocampal seizures. However, all mice populations (Chr2, ArchT or control in SUB) progressed to at least stage 3 seizures with repeated kindling.

Minor Issues

1. Page 22 – “rearing and falling” instead of “rearing and failing”
2. Page 5 – promotor to promoter
3. Page 5 – “...we first created non-specific lesions in the subiculum...” How lesions were done is not described in the method section.
4. Page 4 - “Adopted an adeno associated virus... into the subiculum” is awkward phrasing.
5. Figure S7 change “glutamnergic” to “glutamatergic”. Same in page 5
6. Page 7 – change “optical cannula on the ANT” to “optical cannula in the ANT”
7. Figure 5g and 5h, indicate what the black bars represent.
8. Page 11 – “...monitored fluorescence changes during hippocampal seizures (Figure 5e and f).” No seizures are seen in these panels.
9. Page 25 – What is your definition of fully kindled animals?
10. For all ArchT experiments it would be more appropriate to use term hyperpolarization than inhibition.
11. Page 15 – “...this effect was mediated by HCN channel-based bursting activities.” HCN can control the extent of bursting, but it does not mediate bursting. In brain neurons, the bursting is mediated by Ca²⁺ channels and possibly to some extent persistent Na current.

REVIEWER COMMENTS

Reviewer #1:

General Comments

“The ms “Deconstructing the Diversity of Subicular Circuits in Seizure Generalization of Temporal Lobe Epilepsy” by Fan Fei and collaborators is an extension of their previous study on how the subiculum is a gatekeeper of temporal lobe seizures generalization. Although the role of subiculum in the initiation and propagation of hippocampal seizures has been shown before, the exact cellular mechanisms, as well as downstream propagation of seizures remained elusive. Here they show that the subiculum itself may generate seizures and that subicular heterogeneity should be taken into consideration and uncover the role of subicular neural circuits with different projections. Indeed, their results clearly show that subicular neural circuits differently modulate hippocampal seizures, in particular, anterior thalamus nuclei-projecting neurons bidirectionally modulates seizures through HCN channel associated mechanisms, while Entorhinal Cortex-projecting circuits attenuate seizure strength. Two in vivo models are studied: a rapid hippocampal kindling and hippocampal kainate mouse. The circuits at play in epileptic discharges are labeled and then modulated by opto and chemogenetics.

The data are new and convincing. They shed a new light on the role of the subiculum in temporal lobe epilepsy. The amount of work, of data and the range of techniques used in the paper is impressive. The logic of the paper is very clear and well maintained through the text. Figures show results which often involve complex experimental design in a comprehensive manner. However, I find that text could benefit from some additional information regarding used methods, as well as some additional statistics. I support the paper publication with limited modifications.”

Response: We would like to express our sincere appreciation for your time to review our manuscript and the positive comments on this study. Your comments are all valuable and very helpful for revising and improving our paper, as well as the important guiding significance to our researches. We hope that the following responses according to your comments could address your concerns. In the remainder of this letter, we discuss each of your comments individually along with our corresponding responses.

To facilitate this discussion, we first retype your comments in *italic font* and then present our responses to the comments in blue color.

Comment 1

“The term “deconstructing” in the title is a bit overstated.”

Response 1:

Thanks. Accordingly, we have revised the title of manuscript as “Discrete Subicular Circuits Control Generalization of Hippocampal Seizures”.

Comment 2

“Figure 1. b&c. I did not fully understand the scales of the heat map. Is the duration similar to the left traces? Was the average Ca signal normalized? May the authors distinguish trials for each animal? Was the surface activated during FS vs GS different? Where were the first signals recorded within the subiculum in FS vs GS? g. Was the same activation pattern observed for FS?”

Response 2:

Thank you very much for your comments.

Accordingly, we've added detailed annotations, including timeline in the x-axis and distinguished trials for each animal of the calcium signal in the revised **Figure 1b and c**. The duration of heatmap was matched with the left traces. The average Ca^{2+} signal was shown as $(\Delta F/F) = (F-F_0)/F_0$. The normalized F_0 represented the average value of 10 s baseline signals before kindling stimulation, which was indicated in the method section of the revised manuscript. The first signal recorded within the subiculum may be induced directly by electrical stimulation.

Further, we did not analyze the difference between sub-regional activation pattern during FS and GS, since Gcamp6s expressed in the whole subiculum in all cases. While, the axon-gcamp6s study in the **Figure 5** directly indicates the difference in circuit-specific activation patterns.

Finally, as the number of activated c-Fos was much less in FS than that in GS (**Figure 1d**), we did not observe such region-specific activation pattern after FS (**Figure 1 for Reviewers**). This indicated that such regional activation pattern was more relevant to GS.

Revised Figure 1b and 1c

Figure 1 for Reviewers Subicular c-Fos activation pattern after focal seizures.

Comment 3

“Figure S1. May the authors provide images of the subicular damages? Seizure stage / Racine scale. As far as I have understood it, the Racine scale is used to quantify seizure stages. How can Racine scales values (in Supplementary Fig 1, Fig 1 ...) be decimals and not 1-2-3-4 or 5?”

Response 3:

Thank you very much for these comments.

According to your suggestion, we have added a representative image of subicular lesion in the revised **Figure S1b**.

Further, for the concern of “Racine scales values”, as we repeated at least 2 times in GS expression experiments (3 for Pre, 3 for Light and 2 for Post, please see detailed protocol in **Figure 1m**), the values of

seizure stage were used the mean value of each mouse in statistics and were sometimes decimals. To make it clearer, the specific values were showed in our raw data file, and we've added detailed experimental description in the methods.

Comment 4

“Figure S3. The ability to elicit seizures by only stimulating subicular pyramidal cells is an important result. How did superficial vs deep layers stimulation affect seizure generation? Where some spontaneous seizures recorded after the kindling procedure?”

Response 4:

Thanks for this excellent question. Accordingly, we've done additional experiments to increase the number of cases in direct photoactivation of subicular pyramidal neurons in superficial and deep layers. We found that optogenetic activation of subicular pyramidal neurons in deep layer more sufficiently promoted seizure development, especially in number of stimulations needed to reach GS (Figure S4c and S4d). We did not record any spontaneous seizures after kindling. Thus, we have put this new data in the Figure S4.

Revised Figure S4

Comment 5

“Figure 2. What is the mechanism of the facilitation of anterior thalamus activity on subicular discharges (apart from Ih)? Is it associated with an hippocampal effect? Is it related to a direct effect on pyramidal cells properties? Is it related to reverberating loops between the thalamus and the subiculum? The later hypothesis seems reinforced by the appearance of seizure restart when the thalamus is activated (Fig2a) or by the appearance of a faster termination when the thalamus is inhibited. A way to address this question would be to quantify the activity at the beginning of the seizure in each condition and evaluate if thalamus activity modulation rather affects initial vs later components of the seizure.”

Response 5:

Thank you very much for this constructive comment. Following your suggestion, to test whether thalamus activity affects initial or later components of the seizure, we have analyzed the latency to GS (initial components of GS) during SUB-ANT photoactivation in sGS expression. The results showed that latency to GS was barely affected (**Figure 2 for Reviewers**), while GSD was significantly extended (later components of GS, **Figure 2f**). This indicates SUB-ANT circuit facilitating seizures through modulation of later components of GS by extending seizure termination. Meanwhile, as we found that direct inhibition of ANT pyramidal neurons also retarded hippocampal seizures (**Figure S14**). Thus, we agree with reviewer that facilitation of anterior thalamus activity on subicular discharges might be the reverberating loops between the thalamus and the subiculum. Accordingly, we have added some discussion in the revised manuscript.

Page 17 paragraph 2 in the revised manuscript:

“An important question is how inhibition of ANT terminals affects hippocampal seizures. During SUB-ANT photoactivation in sGS expression, we found that the latency to GS (initial components of GS) was barely affected, while GSD (later components of GS) was significantly extended, which indicated SUB-ANT circuit facilitating seizures through modulation of later components of GS by a longer termination. Such phenomena could be due to the reverberating loops between the thalamus and the subiculum, which needs further elucidation.”

Figure 2 for Reviewers Effect of photoactivation of SUB-ANT circuit on latency to GS during sGS expression.

Comment 6

“May the authors explain in the legend the meaning of AD, AV, AN?? What does that mean?
b: any explanation for the drop happening in the 10 stim both for ADD and sz stage?”

Response 6:

Thank you for these comments. AD, AV and AM refer to anterodorsal thalamic nucleus, anteroventral thalamic nucleus and anteromedial thalamic nucleus, respectively, which were sub-regions of the ANT. We’ve added explanations in the figure legend.

As kindling stimulations were performed 10 times per day, intervals between 10th and 11th, 20th and 21st kindling were separated by about 20 hours, thus the drop of seizure stage and ADD suggested that there was recovery of seizure susceptibility in each animal. This phenomenon was also reported in our previous studies and other groups (Chen et al., 2020, *Nat Commun*, PMID: 32066723; Brandt et al., 2006, *Epilepsia*, PMID: 17116018).

Comment 7

“Figure 3. The anatomical limits of the entorhinal cortex vs subiculum are not clear (3a). May the authors provide an anatomical image in addition the fluorescent one?”

“We still observed some level of cellular damage in subicular pyramidal neurons” (P9): please indicate percentage of cell loss and how it was assessed.”

Response 7:

Thank you very much for your comments. Accordingly, we have revised the representative sagittal image with new added anatomical location in the revised **Figure 3a**.

Further, we have measured cell loss in the subiculum via NeuN staining in chronic period of KA-induced TLE. The subiculum underwent about 39.2% (434.1 ± 20.5 vs. 264.1 ± 32.3 , mean value of three subicular coronal slices of each mouse, 3 mice for each group, $P = 0.0224$) neuronal loss in our KA model (**Figure S9**). Thus, we have added this new data in the revised manuscript as following:

Revised Figure 3a

Revised Figure S9a and S9b

Comment 8

“Figure 4. Since the EC is very close to the subiculum, how can the author validate that CNO did not affect directly subicular neurons. Further, did the authors test the effects of CNO, known to be epileptogenic, in WT KA mice?”

In addition to time-frequency displays, may the authors show examples of seizures in control vs CNO condition?”

Response 8:

Thank you very much for your comments.

The injection site of the EC we chose was Bregma: AP -4.8 mm; L -4.0 mm; V -3.0 mm, which had a certain distance to the subiculum. In our preliminary experiment, we injected 500 nL 0.05% Trypan Blue into the EC of wild-type mouse to determine the exact location of drug diffusion. The result showed this volume of liquid

was expressed strictly in the EC, and did not affect hippocampal areas, including the subiculum (**Figure 3 for Reviewers**), indicating CNO might not directly affect subicular neurons.

Further, we have tested the effects of direct intra-ANT CNO injection on KA-induced chronic seizures in mice with mCherry as suggested, and the results showed that CNO itself had no influence on the number and duration of FSs or GSs. These data were presented in the revised **Figure S9c-f**.

Finally, according to your suggestion, we have showed the corresponding seizure examples of the time-frequency displays in the revised **Figure 4h and m**.

Figure 3 for Reviewers Exact location of Trypan Blue injection in the EC

Revised supplementary figure 9c-f

Revised Figure 4h

Revised Figure 4m

Comment 9

“Figure 5.

Page 10: “We observed that the somatic locations of the ANT-projecting neurons and the EC-projecting neurons in the subiculum were totally segregated. Specifically, ANT-projecting neurons were almost completely deep located while EC-projecting neurons were mostly distributed in the superficial layer. Such topological distribution was consistent across the anterior-posterior axis (Figure 5c and d)”. It’s unclear from the text whether this was shown before or it’s an origin finding of this study.”

Response 9:

Thank you very much for this important comment. Indeed, subicular projections to cortical and subcortical regions showed heterogeneity were reported before (Witter et al., 1990, PMID: 12106290; Aggleton and Brown, 1999, PMID: 11301518, etc.). Yet no existing studies directly distinguished the specific locations of ANT-projecting and EC-projecting subicular pyramidal neurons as far as we know. To make it clearer, we have indicated our origin finding in the result description as following:

Page 11 paragraph 2 in the revised manuscript:

“These results indicated that most subicular neurons projecting to distinct downstream areas were not collateral projections but had structural heterogeneity. Although heterogeneity in subicular projections to cortical and subcortical regions was reported before, yet no existing studies directly distinguished the specific locations of ANT-projecting and EC-projecting subicular pyramidal neurons.”

Comment 10

“Figure 6.

h. May the authors display a picture of the fluorescence pattern after CTB injection in the anterior thalamus (rather than a cartoon).

i. The color code is disturbing since red here stands for bursting while it is used for cells connected to the ANT in the rest of the figure. Please change.”

Response 10:

Thank you very much for your suggestion. We’ve showed a picture of the fluorescence in the revised **Figure 6h** and also changed the color of the revised **Figure 6i**.

Revised Figure 6h and 6i

Comment 11

“Figure 7e. The 2 last FP examples are not aligned with the trace.
Does the HCN activator antiepileptic drug Lamotrigine worsen seizure spread?”

Response 11:

Thank you very much for your comments.

We’ve corrected it in the revised Figure 7e.

Further, following your suggestion, we’ve tested the effect of intra-subicular LTG injection on the GS expression. LTG at 100, 200 μM doses (intra-subicular, 500 nL, the same for 500 μM) did not promote or dampen seizure stage, ADD or GSD, while LTG at 500 μM dose had a tendency to alleviate seizure severity (Figure 4 for Reviewers). Meanwhile, we also intraperitoneally injected LTG (20 mg/kg) to verify its anti-seizure effectiveness (Figure 4 for Reviewers). This can be due the fact that LTG is also a strong sodium and calcium channel blocker, e.g., as LTG at 50 μM *in vitro* significantly inhibited Na^+ current (Rogawski and Loscher, *Nat Rev Neurosci*, 2004, PMID: 15208697). To avoid complicating the result interpretation, we did not include these data in the revised manuscript.

Revised Figure 7e

Figure 4 for Reviewers Effects of intra-subicular lamotrigine injection on kindling-induced sGS.

Comment 12

“Methods.

Page 21: “We only included mice with correct electrode, cannula emplacement, and viral expression”. It would be nice to mention the percentage of mice which were excluded from analysis based on these criteria, in order to show the feasibility of these experiments.

Page 21: “Only mice with detectable seizures were given CNO injections”. Please indicate the percentage of mice with detectable seizures.”

Response 12:

Thanks for these questions. The percentage of mice with corrected correct electrode, cannula emplacement, and viral expression the was about 70% (As an example 31/44 mice used in Figure 1i-l). Besides, due to the

mortality of KA model, the ratio of animals with detectable seizures during recording period versus animals injected with KA was about 40-50% (9/21 mice used in SUB-ANT circuit and 6/15 mice used in SUB-EC circuit). To make it clearer, we've both indicated these details in the revised manuscript as following:

Page 22 paragraph 3 in the revised manuscript:

"We only included mice with correct electrode, cannula emplacement, and viral expression (approximately 30% mice were excluded under these criteria)."

Page 24 paragraph 1 in the revised manuscript:

"Only mice with detectable seizures were given CNO injections (The percentage of mice with detectable seizures in the study was about 40-50%)."

Comment 13

"Page 23: "Recorded putative pyramidal neurons were post positioned for their sub-regional locations in the SUBa (the deep part of the subiculum) or SUBb (the superficial part of the subiculum)". Please clarify in the methods how neuronal positioning was verified."

Response 13:

Thank you very much for this comment. We defined the neuronal position with the following criteria: when electrodes were positioned at AP -3.3 to -3.5 mm, neurons at depths of -1.30 to -1.65 mm were located in SUBa and neurons at depths of -1.65 to -1.90 mm were located in SUBb; when electrodes were positioned at AP -3.5 to -3.7 mm, neurons at depths of -1.70 to -1.80 mm were located in SUBa and neurons at depths of -1.80 to -2.00 mm were located in SUBb; the rest that were not in these locations or whose locations could not be identified were excluded. To make it clearer, we've indicated detailed method description in the revised manuscript. Meanwhile, we've also showed approximate electrode placements of in vivo multi-unit recordings after post-verification in the revised **Figure S11**.

Revised Figure S11

Comment 14

"Page 24 - 3.3 mM CaCl₂. Why calcium concentration in ACSF was so high and what was the composition of ACSF in which recordings were performed? If the same as storage ACSF, it should be specified."

Response 14:

Thank you very much for this comment. Actually, it was a miscalculation as we weighed 367.5 mg CaCl₂·2H₂O but used the mole mass of CaCl₂ in conversion. The exact concentration should be 2.5 mM. We've corrected in the revised manuscript. We've also indicated in the revised manuscript that recording ACSF was the same as storage ACSF.

Comment 15**Minor.**

"Page 3. "which is the common origin of epileptic seizures during TLE8". I am not sure that the reference (on comorbidities) is the good one."

Response 15:

Thank you very much. We've changed a more suitable reference instead as follows:

Thom, M. Review: Hippocampal sclerosis in epilepsy: a neuropathology review. 2014, Neuropathol Appl Neurobiol.

Comment 16

"Page 4: "An intrinsic enhanced Ih-dependent burst intensity, underlying the synaptic plasticity of ANT-projecting subicular pyramidal neurons, facilitates the generalization of hippocampal seizures, which may be of therapeutic significance due to an improved understanding of the subicular neural circuit in TLE." probably should be facilitates"

Response 16:

Thank you very much. We've corrected the typo.

Comment 17

"Figure 1b. Kindling stimulatoin should be corrected."

Response 17:

Thank you. We've corrected the typo of 'stimulation' in the revised **Figure1b**.

Comment 18

"Fig 5g & j : please add that this relates to FS"

Response 18:

Thank you. We've added annotations in the revised **Figure 5**.

Revised Figure 5g-n

Comment 19

“Page 14: “The subiculum is thought to be essential in the initiation and propagation of seizures in epilepsy, but how long-projecting subicular pyramidal neurons and related circuits were recruited was unknownrarely revealed before”. “unknownrarely revealed before” should be rephrased.”

Response 19:

Thank you. We’ve rephrased the sentence as “how long-projecting subicular pyramidal neurons and related circuits recruited was not revealed before”.

Reviewer #2:

General Comments

“The current study explores the role of principal cells from the subiculum in temporal lobe epilepsy (TLE). Using two different models of TLE (kindling and kainate) combined with optogenetics, chemogenetics, calcium imaging, tracing and electrophysiology, the authors show that the stimulation of pyramidal neurons projecting to the anterior nucleus of the thalamus (ANT) promote epileptic seizures while their inhibition reduced the severity of the seizures. By contrast, the selective silencing of pyramidal cells projecting to the entorhinal cortex (EC) facilitated seizures. ANT projecting neurons express a hyperpolarization-activated current (I_h) and are capable of firing action potentials in bursts. Knocking down the expression of I_h alleviates the development of seizure.

The topic is important; the results are interesting and most of them are well supported by an impressive range of state-of-the-art techniques. However, I have some concerns.”

Response: We would like to express our sincere appreciation for your time to review our manuscript and the positive comments on this study. Your comments are all valuable and very helpful for revising and improving our paper, as well as the important guiding significance to our researches. We hope that the following responses according to your comments could address your concerns. In the remainder of this letter, we discuss each of your comments individually along with our corresponding responses.

To facilitate this discussion, we first retype your comments in *italic font* and then present our responses to the comments in blue color.

Specific comments

Comment 1

“Why did not the authors test the effect of an activation of EC, NAc and MMB projecting neurons (as they did for ANT projecting neurons)?”

Response 1:

Thank you very much for your constructive comment. Compared with optogenetic inhibition experiments, optogenetic activation of projecting terminals might have a risk to retrogradely activate collateral-projecting circuit, thus it may not circuit-specific (Yizhar et al., *Neuron*, 2017, PMID: 21745635). Thus, in our original version of manuscript, we did not give priority to testing the effect of an activation of EC-, NAc- and MMB-projecting neurons. Following your comment, we have also performed additional experiments to test the effect of optical activation of SUB-EC, -NAc and -MMB circuits in seizure modulation. We found that activation of these three circuits had no significant effects on kindling-induced seizures (**Figure S8**), further suggesting the importance of SUB-ANT circuit in seizure modulation. Thus, we have added this new data in the **Figure S8** of revised manuscript.

Figure S8. Effects of optogenetic activation of SUB-EC, SUB-NAc and SUB-MMB circuits on hippocampal kindling model.

Comment 2

“- Fig 5k What does the Ca increase in EC terminal visible before GS onset correspond to?”

Response 2:

Thank you for your question. The increased period of Ca²⁺ signal was corresponding to behavioral latency to GS. This suggests that transient excitation in SUB-EC terminal might be involved in seizure initiation, which may be explained by previous study that the initial progression of the ictal discharge from SUB/CA1-EC (Gnatkovsky et al., *Ann Neurol*, 2008, PMID: 19107991) or the SUB-EC interaction during ictogenesis (Herrington et al., *Seizure*, 2015, PMID: 26362375). To make it clearer, we have indicated this phenomenon in the revised manuscript as following:

Page 12 paragraph 1 in the revised manuscript:

“During sGSs (Figure 5j), fluorescence in the ANT terminal showed a delayed increase with seizure development (Figure 5k and l); in the EC terminal, fluorescence temporarily increased (corresponding to behavioral latency to GS) but then decreased even below the baseline, which lasted long after sGS termination (Figure 5m and n).”

Comment 3

“Optogenetic activation and inhibition of subiculum principal cells: the authors should document the response to blue light at cellular level, ideally by patching neurons. They should provide evidence that the stimulation of Arch in the projecting regions of subicular neurons inhibits neurotransmitter release and does not revert the

effect of presynaptic GABA receptors due to the accumulation of Cl⁻ in the intracellular space (see PMID 22729174).”

Response 3:

Thank you very much for your comment.

Following your suggestion, we have validated that subicular pyramidal neurons expressing ChR2 responded to 20 Hz 473 nm light stimulation (7/9 neurons from 3 mice) *in vitro* electrophysiology. Evoked APs lasted for a while even after the end of light stimulation (**Figure S2a and S2b**).

Meanwhile, as Arch hyperpolarizes neurons through outward H⁺ transportation, basically it will not change the concentration of Cl⁻ and revert the effect of presynaptic GABA receptors. Indeed, dysregulation of Cl⁻ concentration in subicular pyramidal neurons contributes to GS. We previously reported that activation of subicular pyramidal neurons genetically targeted with Arch, but not NpHR3.0, alleviated GS expression (Wang et al., *Neuron*, 2017, PMID: 28648501). Therefore, in the present study, we used Arch to optogenetically inhibited subicular PNs and projecting terminals. Following your comment, we have also validated that 589 nm light stimulation of subicular PNs expressing Arch significantly decreased EPSC frequency in ANT pyramidal neurons, and did not cause rebound activity (6 neurons from 3 mice, **Figure S7**).

Thus, we have added new data in the revised manuscript as following:

Revised Figure S2a and S2b

Revised Figure S7

Comment 4

- Slice electrophysiology: The burst firing examples in fig 6l and 6n seem to be superimposed on a low-threshold calcium spike produced by a T-type calcium current. T-channels (CaV3) are important for burst firing in the subiculum (e.g., PMID: 28744923; 28326015). In addition, it was shown that the compound used in the study for blocking I_h (ZD7288) is not selective as it also blocks T-type calcium current (PMID: 18534748). The authors should test for the possible contribution of CaV3 channels to the bursting behavior and to the induction of epileptic seizures.

Response 4:

Thank you very much for your constructive suggestion. Following your suggestion, we thus have tested effects of T-channel blocker, TTA-P2 on bursting firing of subicular pyramidal neuron and hippocampal seizures. We found that TTA-P2 successfully inhibited bursting intensity *in vitro* without affecting hyperpolarizing current or RMP (Figure S13a-d), and TTA-P2 also inhibited seizure stage and GSD in kindling-induced GS expression (Figure S13 e and f). This suggested that HCN and T-channels might synergistically contribute to the extent of pyramidal neuron bursting and inhibited both of them have anti-seizure effect. Thus, we've added this new data and some discussion in the revised manuscript.

Revised Figure S13

Page 14 paragraph 2 in the revised manuscript:

“In addition, as previous studies showed that T-type calcium channel contributed to bursting of subicular pyramidal neurons 40, we here also confirmed this (Figure S13a-d) and further found that intra-subicular injection of T-type calcium channel blocker, TTA-P2 significantly reduced seizure stage and shortened GSD in sGS expression as well (Figure S13e and f). These results confirmed that bursting firing of pyramidal neurons in the subiculum was important in generalization of hippocampal seizures.”

Page 20 paragraph 1 in the revised manuscript:

“In addition, T-type Ca²⁺ channel in the subiculum is also an important factor contributing to pyramidal neuron bursting 40. Pharmacological blockage of T-type Ca²⁺ channel significantly inhibited bursting intensity of subicular pyramidal neurons and hippocampal seizures, further confirming that bursting firing.”

Comment 5

“- In favor of a contribution of Ih, the authors show that silencing the expression of HCN1 in ANT projecting neurons. However, Ih contributes to the resting membrane potential of several neurons (e.g., PMID 9644226; 18216232) and silencing it could result in a hyperpolarization of pyramidal neurons which would also inhibit the activation of ANT. To validate their hypothesis, the authors should show that silencing HCN is sufficient to eliminate burst firing.”

Response 5:

Thank you very much for your suggestion. Following your comment, we've performed experiments to test the effect of HCN KD on subicular pyramidal neuron bursting. Indeed, knocking down of HCN1 hyperpolarized the RMP, and also decreased AP number on burst in ANT-projecting subicular PNs (Figure S12). This further

confirmed HCN-associated mechanism in bursting feature in SUB-ANT circuit. Thus, we have added this new data in the revised **Figure S12**.

Revised Figure S12

Minor comments

Comment 6

“Fig. 1b: typo”

Response 6:

Thank you. We’ve corrected the typo of ‘stimulation’ in the revised **Figure 1b**.

Comment 7

“p. 6, first paragraph: “the pro-seizure effect of optogenetic activation of subicular pyramidal neurons was reflected by the increasing number of stimulations needed...” I guess the authors mean the opposite, as shown in fig 1jk.”

Response 7:

Thank you. We’ve rephrased the sentence as “while the pro-seizure effect of optogenetic activation of subicular pyramidal neurons was reflected by the decreasing number of stimulations needed to reach GS” in the revised manuscript.

Reviewer #3:

General Comments

“Fei et al. investigated the role of excitatory subicular neurons in the generation of epileptiform activities. By primarily using the kindling model of epilepsy in mice, the authors measured calcium signaling in the subiculum during seizures. They manipulated the excitability of the subiculum or its downstream terminals using optogenetics or chemogenetic tools targeted to excitatory neurons. The major strength of the study is the use of a variety of methods trying to answer the same question. The data show:

- 1. Increased calcium signalling in excitatory subicular neurons at the onset of electrographic seizures.*
- 2. Behavioural seizures are more severe when subicular pyramidal neurons are activated optogenetically and less severe when hyperpolarized.*
- 3. Optogenetic activation of excitatory, subicular terminals in the anterior thalamus (SUB-ANT) worsens seizure severity while hyperpolarization of SUB-ANT lowers seizure severity.*
- 4. SUB-ANT projections originate in laterodorsal subiculum and subiculo-entorhinal (SUB-EC) projections originate in posteroventral subiculum. Calcium response during seizures reaches its maximum earlier in SUB-EC neurons than in SUB-ANT neurons although the signal in SUB-EC neurons is weaker.*

Merits

In studying subicular outputs, the authors have identified an important pathway in epileptogenesis and have performed extensive experiments to investigate the underlying mechanisms. The experiments are generally well-designed to address the hypotheses outlined in the text. Although many unclear aspects of results and their description are present in the manuscript, the main conclusion will likely stay unchanged: subicular projections to anterior thalamus boost seizure and subicular projections to entorhinal cortex dampen seizures.”

Response: We would like to express our sincere appreciation for your time to review our manuscript and the positive comments on this study. Your comments are all valuable and very helpful for revising and improving our paper, as well as the important guiding significance to our researches. We hope that the following responses according to your comments could address your concerns. In the remainder of this letter, we discuss each of your comments individually along with our corresponding responses.

To facilitate this discussion, we first retype your comments in *italic font* and then present our responses to the comments in blue color.

Major critique.

Comment 1

“1. The title is misleading. According to International League Against Epilepsy, epilepsy is characterized by unprovoked seizures. The main method used here to trigger seizures is electrical kindling, which is by definition stimulation triggering paroxysmal response and it is not spontaneous.”

Response 1:

Thank you for your comment. We totally agree with your opinion. Accordingly, we have revised the title of manuscript as “Discrete Subicular Circuits Control Generalization of Hippocampal Seizures”.

Comment 2

“2. Electrophysiological part of the study is either not properly done or not properly described. No reliable conclusions can be made from this part of the study (details are below).”

Response 2:

Thank you very much for your comment. As suggested, we've carefully checked our electrophysiological studies and provided detailed descriptions and results in the revised manuscript. All the revisions are highlighted by yellow color. Please also see the following part in details.

Comment 3

"3. Separation of focal and generalized seizures is misleading. First, FS are described as stage1-3 on Racine scale, then as paroxysmal event lasting 10-30 sec. According to accepted definition, focal seizures occupy just one area of the brain and generalized seizures bilateral and occupy multiple brain areas. In the study, there was no even attempt to obtain data enabling to characterize seizures as either focal or generalized. There is also a terminology like hippocampal seizures that is used in conjunction with generalized seizures. Hippocampal seizures cannot be generalized. Generalized seizures can be primarily hippocampal sans secondary generalised."

Response 3:

Thank you for your comments. We totally agree that the term 'GS' we used in original version of manuscript actually is hippocampal focal seizure with secondarily generalized seizure according to many previous studies (Blumenfeld et al., *Brain*, 2009, PMID: 19339252; Caciagli et al., *Neurology*, 2020, PMID: 32847951). Thus, we've rephrased 'generalized seizures' as 'secondary generalized seizures' instead throughout the revised manuscript.

Comment 4

"4. Controls are often absent."

Response 4:

Thank you for your comment. We've done additional control experiments as suggested. Please also see the following part in details with new added control groups (Figure S2c for optogenetic tools in kindling model, figure S9c-f for chemogenetic tools in KA model)

Comment 5

"5. It was often impossible to know how many animals, slices etc were used for the analysis."

Response 5:

Thank you for your comment. We have carefully checked all figures and indicated all number of cases used in figures or figure legends.

Comment 6

"6. Method section does not give sufficient details, therefore often, it was impossible to evaluate accuracy of provided results."

Response 6:

Thank you for your comment. We've provided detailed descriptions about all experiment produces in the revised manuscript. All the revisions are highlighted by yellow color. Please also see the following part in details.

Specific critique.

Comment 7

“1. As presented, the results are convincing in demonstrating the link between subiculo-thalamic output and seizure severity. However, the lack of detail in the methods section casts doubt on the validity of the results.”

For example:

a. It is unclear how seizure scoring was performing. The authors state that they used the Racine scale but there is no detail as to the criteria used to detect behaviour such as facial movements, myoclonic jerks etc. Was the scoring performed automatically? How many facial movements constituted stage 1? If done manually, was the testing blinded and/or randomized?”

Response 7:

Thank you for the detailed comments. In the present study, seizure stage was scored manually by well-trained experimenters who are blinded to the group allocation. As long as the mouse exhibited one myoclonic facial movement, we identified it as stage 1. The protocol is consistent with our previous studies and other groups (Chen et al., 2020, *Nat Commun*, PMID: 32066723; Brandt et al., 2006, *Epilepsia*, PMID: 17116018). To make it clearer, we've indicated in the method section of the revised manuscript.

Comment 8

“b. How were after discharge durations calculated? What constituted a single-event? For example, in Figure 1L: eYFP-GS, do we see one long ADD or two separate ADDs according to the authors' criteria? In Figure 4c, what is shown as a focal seizure is indistinguishable from theta activity which we expect in hippocampal recordings.”

Response 8:

Thank you for your question. The length of ADD was defined as the duration between the moment of kindling stimulation and the end of paroxysmal discharge event in EEG. It is actually a single seizure rather than two separate ADDs in **Figure 1L**. The amplification of the sample EEG is shown in **Figure 5 for Reviewers** as following. In addition, we have also added the amplification of the sample base EEG section in the revised **Figure 4C**, which differed from theta activity in basal hippocampal EEG.

Figure 5 for Reviewers Sample GS EEG in the Figure1L

Revised Figure 4c

Comment 9

“c. Was kindling performed only during wakefulness? Sleep is a strong factor influencing the expression of seizures and it constitutes a major confound.”

Response 9:

Thank you for your question. Kindling in our study was performed only during wakefulness, and we’ve indicated in the method section of the revised manuscript.

Comment 10

“d. Which AAV serotype was used for tracing and optogenetic experiments? Most AAVs have retrograde properties. EC is reciprocally connected with the subiculum such that AAV injections in the subiculum can be taken up by EC terminals and transported retrogradely to the EC (i.e., we this somewhat in FigS4 with puncta in EC and RSG indicating cell bodies). Thus optogenetic stimulation in EC would modulate subicular neurons, possibly explaining the contrary results obtained from EC stimulation.”

Response 10:

Thank you for your questions. The AAV serotypes used in present study were AAV2/8 and AAV2/9, which are all indicated in the method section of the revised manuscript. To make it clearer, we further showed high-resolution images in the **revised Figure S5**. Indeed, no fluorescence was seen in the somata of EC neurons in the enlarged view (Scale bar, 200 μ m, 10 μ m, respectively, **Figure 6 for Reviewers**).

Revised Figure S5

Figure 6 for Reviewers Subicular glutamatergic projecting terminal in the EC

Comment 11

“e. KA model of epileptogenesis is not described.”

Response 11:

Thank you for your comment. We’ve added detailed descriptions about KA model of epileptogenesis in the method part of the revised manuscript.

Page 24 paragraph 3 in the revised manuscript:

“In this model, SE is typically induced via intra-hippocampal injection of KA (as mentioned in the **Stereotactic injections and surgeries** section), which further caused spontaneous recurrent seizures in the following several months.”

Comment 12

“f. How was cFos counted? How many animals? Slices? Slices per animal? If only one slice, how this slice was chosen?”

Response 12:

Thank you for your questions. We counted the number of c-Fos⁺ cells using following protocol: we calculated the mean value from three representative coronal slices (anterior, intermediate and posterior, e.g., for the subiculum, approximately AP -3.0, -3.4 and -3.8 mm) in each mouse, and the number of mice used for group statistics was indicated in each figure (at least $n \geq 3$). To make it clearer, we've added this detailed description in the revised manuscript.

Page 29 paragraph 3 in the revised manuscript:

“For c-Fos⁺ and NeuN⁺ cell quantification, we counted the number of cells that exhibited c-Fos⁺ or NeuN⁺ immunoactivity within the corresponding nucleus of three representative coronal slices (anterior, intermediate and posterior), and then calculated the mean value for each mouse. The number of mice used in each experiment was indicated in figure legends.”

Comment 13

“g. c-Fos signaling degrades fast. What was the time between seizure and euthanasia for C-fos experiments?”

Response 13:

Thank you for your question. As reported, c-Fos protein expression was the highest at 1.5 h time point (Xiu et al., *Nat Neurosci*, 2014, PMID: 25242305). We thus transcranially perfused mice 1.5 h after seizure termination. To make it clearer, we've added the detailed description in both the result and method sections of the manuscript.

Page 4 paragraph 3 in the revised manuscript:

“We analyzed expression of the immediate early gene Fos in the subiculum 1.5 h after seizures.”

Page 29 paragraph 2 in the revised manuscript:

“Notably, in c-Fos staining experiments, animals were sacrificed 1.5 hours after seizures.”

Comment 14

“h. How was power calculated in Figure 2g?”

Response 14:

Thank you for your question. The EEG power recorded was calculated and analyzed offline by a software package (Scan 4.5) in the Neuronscan System, and we've indicated it in the method section of the revised manuscript.

Comment 15

“i. Chemogenetic experiments seems do not have any controls. Was there CNO injection without virus tested? Was there saline injection in animals with viruses expressed in target structures?”

Without the above details, the validity of the data is impossible to judge.

Response 15:

Thank you for your questions. As suggested, we've added experiments to test the effects of intra-ANT CNO injection on KA-induced seizures in *CaMKII α -mCherry* mice. The results showed that CNO had no influence on the number and duration of FSs or GSs (**Figure S9c-f**). Besides, in 'Pre' and 'Post' periods, we injected saline before recording started. We found no difference in times and durations of FSs among each day during 'Pre' period in Figure 4 (ANT n=8 and EC n=6, **Figure 7 for Reviewers**), indicating that saline may not affected seizure severity. Thus, we have added this new data in the revised **Figure S9c-f**.

Revised Figure S9c-f

Figure 7 for Reviewers Number and Time of FS during Pre in KA-induced chronic seizures

Comment 16

"2. Please include low-magnification histological sections both in the main body and supplemental figures along with the current pictures. It is hard to determine where injections were made without seeing anatomical landmarks."

Response 16:

Thank you very much for this comment. Following your suggestion, we've provided low-magnification histological sections and labelled some common anatomical landmarks in the revised histological images to make it clearer. One representative image is showed as following:

Revised Figure 3a

Comment 17

“3. The separation of the ictal broadband into alpha, beta, theta, delta and gamma is not appropriate. These rhythms are activity and state-dependent and are likely not all present within the seizure event. The result may reflect change in the overall power of the signal rather than a change in the power of individual frequency bands.”

Response 17:

Thank you for your comments. As suggested, we’ve calculated the overall powers before and after light stimulation in optogenetic studies. We found that photoactivation of SUB-ANT circuit increased total EEG power (0-100 Hz) in secondary GS period, while hyperpolarizing of SUB-ANT circuit decreased total EEG power. Thus, these results were added into the revised **Figure2 g and 2o**.

Revised Figure 2g and 2o

Comment 18

“4. Page 6- “We found that activation of the subicular pyramidal neurons prolonged GS durations (GSDs), leaving seizure stages and ADDs unaffected (Figure 1n-p). In contrast, inactivation of subicular pyramidal neurons reduced GSDs (Figure 1q-s).” Did you do any controls here?”

Response 18:

Thank you very much for this comment. In the original version of manuscript, we used self-control group to test the effects of activation of the subicular pyramidal neurons on sGS. Following your question, we have also performed additional experiments to test the effects of light stimulation (20 Hz 473 nm) in the subiculum of *CaMKII*-eGFP mice on sGS. We found that light itself had no influence on sGS expression (Figure S2c). Thus, we have added new data in the revised **Figure S2c** as following:

Revised Figure S2c

Comment 19

“5. Fig. S4. I do not certainly understand what we see in B. If this is anterograde tracing, we should see only axons in target structures. The signal in most of the cases is so strong, that it look like the whole structure is composed of exclusively subicular axons, which cannot be true.”

Response 19:

Thank you for your comments. In the original version of manuscript, the strong signal in the axons might be due to the large amount of viral infection and long exposure time in imaging. Following your comment, we’ve redone the virus tracing experiment with less AAV injection volume (100 nL instead of 200 nL) in the subiculum. The results were shown in the revised **Figure S5**. Fluorescence in the downstream regions was truly subicular projecting terminals, and was consistent with to some other anatomical studies about output of the subiculum (Roy et al., *Cell*, 2017, PMID: 29161563; Kitanishi et al., *Sci Adv*, 2021, PMID: 33692111).

Revised Figure S5

Comment 20

“6. Fig. S5 and all related text. Was there any attempt for quantification of data? Fig.S4 suggests that there are very strong projections from Subiculum to LD thalamic nucleus. Despite this fact, nothing in the study addresses a role of LD and here, it would be important to see at least cFos expression in LD?”

Response 20:

Thank you very much for this suggestion. Accordingly, we've added experiments to stain c-Fos expression in the LD after GS in the revised **Figure S6d**. Meanwhile, we've also quantified the c-Fos expression in the main subicular downstream regions in the revised **Figure S6h**. We found that c-Fos expression in the LD was much less than other nuclei like the ANT, EC, NAc and MMB. It is possible that heterogeneously distributed GABAergic neurons in the dorsal thalamus (Arcelli et al., *Brain Res Bull*, 1997, PMID: 8978932) might recruit diverse local inhibitory microcircuits, thus limited c-Fos expression in the LD. Thus, we have added this new data in the revised **Figure S6**.

Revised Figure S6

Comment 21

"7. The author state that they recorded 'single unit activities'. This is not true. At maximum, they recorded multiunit activities with post processing aiming to obtain information on single unit firing. Definitely, in the type of experiments done here, it is impossible to obtain information on single unit firing derived from multiunit recordings. As described by 'fathers' of spike sorting (Harris KD, Henze DA, Csicsvari J, Hirase H, Buzsaki G (2000) *J Neurophysiol* 84 (1):401-414) the major errors of soring come (a) in bursting neurons, because the amplitude and shape of every spike produced by the same cell is different and (b) the errors of detection are high when neurons fire synchronously, like in your study, during paroxysmal discharges. Thus, Spike sorting procedures cannot give any reliable information when applied to bursting neurons that tend to fire synchronously."

Response 21:

Thank you very much for this important comment. Indeed, we recorded multiunit activities with post processing aiming to obtain information on single unit firing. We totally agree with you that errors of spike sorting may come when neuron firing synchronously or in a bursting state. Thus, we've revised "multi-unit activities" instead of "single unit activities".

Meanwhile, since *in vivo* recording lacked the bursting precision of SUB-ANT circuits, we next introduced CTB-based slice electrophysiology (Figure 6h). As a result, the ANT-projecting bursting neurons showed a higher burst firing intensity, revealed by more spikes and shorter spike intervals in their bursting action potential (Figure 6l and m).

Comment 22

“8. Neuronal firing activities were recorded with 12 microwire bundles combined with optical fiber. If one assumes the use of a glue or epoxy to keep all this together, the object inserted into the brain is in the order of 1 mm, which typically produces major damage to the brain including cutting all passing fibers.”

Response 22:

Thank you very much for your comment. We apologize for unclear description in the original method section. Indeed, 12 microwires were twisted into a bundle. The diameter of this cylinder-like recording electrode was about 0.1 mm (4 times single microwire’s diameter), and the diameter of optical fiber was 0.2 mm. Therefore, the total diameter of the object was about 0.3 mm. Representative image showed that there is only minor damage to the brain (Figure 6a). To make it clearer, we’ve added detailed descriptions for recording electrodes in the method of the revised manuscript.

Page 25 paragraph 2 in the revised manuscript:

“12 microelectrodes were twisted into a bundle to form recording electrodes, which had an impedance of 1-2MΩ and were combined with an optical fiber terminal to keep stiff (the diameter was about 0.3 mm)”.

Comment 23

“9. Classification of neurons with increased/decreased activities with a criterion 20% is not acceptable. Statistical tests are regularly done in other labs to evaluate either increase or decrease in firing.”

Response 23:

Thank you very much for this constructive comment. According to statistical test used in the previous study (Chen et al., 2020, *Nat Commun*, PMID: 32066723), we’ve changed the criterion as ‘neurons that increased >2 SDs of baseline average were defined as ‘increase’, neurons that decreased >2 SDs of baseline average were defined as ‘decrease’, the rest were defined as ‘no change’. Under this new criterion, we’ve re-analyzed the statistics of multiunit recordings. The main conclusion drawn from this figure was unchanged, that is pyramidal neurons in the SUBa and SUBb showed functional heterogeneity during hippocampal seizures. Thus, the result was shown in the revised Figure 6c.

Revised Figure 6c

Comment 24

“10. The use of *in vitro* electrophysiological recordings in this study goes below any critique. The bath solution here contained 2.5 KCl and 3.3 CaCl₂, which is more less typically used *in vitro*. However, during seizures the extracellular K goes to 12-16 mM and extracellular Ca to 0.6 mM. Needless to say, that Ca is major ion mediating bursting in pyramidal cells and K one of the major ions contributing to Ih. In addition, the membrane potential is not indicated making impossible to judge whether observed changes were due to specific action of ZD7288 or associated this it changes in the membrane potential.”

Response 24:

Thank you for pointing out this important issue. Actually, the exact concentration of Ca²⁺ was 2.5 mM after careful check (As we weighed 367.5 mg CaCl₂·2H₂O but used the mole mass of CaCl₂ in conversion). Thus, we’ve corrected it in the revised manuscript.

In addition, ZD7288 slightly hyperpolarized the RMP of ANT-projecting subicular pyramidal neurons, and we’ve included in the revised **Figure 6p**. Indeed, in addition to Ih, subicular T-type Ca²⁺ channel is important in mediating pyramidal neuron bursting. We have thus tested the T-type Ca²⁺ channel blocker, TTA-P2 on subicular neuronal bursting and seizures. We found that TTA-P2 successfully inhibited bursting intensity *in vitro* without affecting hyperpolarizing current or RMP (**Figure S13a-d**), and it also inhibited seizure stage and GSD in kindling-induced sGS (**Figure S13e and f**). Thus, we suggested HCN and T-channels synergistically contribute to the extent of pyramidal neuron bursting and inhibited both of them have anti-seizure effect. Accordingly, we’ve added these new data and some discussion in the revised manuscript.

Revised figure 6o and 6p

Revised Figure S13

Comment 25

“11. Page 11: “we subsequently defined the location of each neuron in the deep or superficial with post-verification”. Please include these data.”

Response 25:

Thank you very much for this comment. We defined the neuronal position with the following criteria: when electrodes were positioned at AP -3.3 to -3.5 mm, neurons at depths of -1.30 to -1.65 mm were located in SUBa and neurons at depths of -1.65 to -1.90 mm were located in SUBb; when electrodes were positioned at AP -3.5 to -3.7 mm, neurons at depths of -1.70 to -1.80 mm were located in SUBa and neurons at depths of -1.80 to -2.00 mm were located in SUBb; the rest that were not in these locations or whose locations could not be identified were excluded. To make it clearer, we’ve indicated detailed method description in the revised manuscript. Meanwhile, we’ve also showed approximate electrode placements of in vivo multi-unit recordings after post-verification in the revised **Figure S11**.

Revised Figure S11

Comment 26

“12. Page 12. “Since in vivo recording lacked the precision of SUB-ANT or SUB-EC circuits, ...”. This is your fault. You could easily use antidromic stimulation in order to identify projecting neurons.”

Response 26:

Thank you very much for your critique. We apologize for the rough conclusion drawn from in vivo multi-unit recordings. CTB-assisted in vitro recordings we’ve done in the revised **Figure 6h-p** might precisely distinguish the properties of circuit-specific subicular neurons. To avoid any confusing, we have rephrased this sentence as following:

Page 13 paragraph 2 in the revised manuscript:

“To more precisely identify and characterize circuit-specific subicular neurons, we next introduced CTB-based slice electrophysiology (Figure 6h).”

Comment 27

“13. Page 12. “...reflecting enhanced function of the somatic hyperpolarization-activated cyclic nucleotide-gated cation (HCN) channel.” What means enhanced function of a channel? The channel can be either open or closed.”

Response 27:

Thank you very much for this comment. We've rephrased the sentence as "These electrophysiological properties were mediated by somatic hyperpolarization-activated cyclic nucleotide-gated cation (HCN) channel".

Comment 28

"14. Page 12, Fig. 6 I, n. "...ANT-projecting bursting neurons showed a higher burst firing intensity..." If the resting membrane potential is similar in shown traces, there is obviously different extent or type of high-threshold Ca channels that mediate difference in burst properties."

Response 28:

Thank you very much for your comment. Indeed, the RMP of ANT-projecting was hyperpolarized after ZD7288 blockage of subicular HCN channels (Revised **Figure 6o and 6p**) or shRNA knockdown of subicular HCN1 (**Revised Figure S12a-c**).

In addition to I_h , subicular T-type Ca^{2+} channel is important in mediating pyramidal neuron bursting as suggested by Reviewer 2. We have thus tested the T-type Ca^{2+} channel blocker, TTA-P2 on subicular neuronal bursting and seizures. We found that TTA-P2 successfully inhibited bursting intensity *in vitro* without affecting hyperpolarizing current or RMP (**Figure S13a-d**), and it also inhibited seizure stage and GSD in kindling-induced sGS (**Figure S13e and f**). Thus, we suggested HCN and T-channels synergistically contribute to the extent of pyramidal neuron bursting and inhibited both of them have anti-seizure effect. Accordingly, we've added these new data and some discussion in the revised manuscript.

Revised figure 6o and p

Revised Figure S12a-c

Revised Figure S13

Comment 29

“15. Experiments with ZD7288 injection. Was there any control done?”

Response 29:

Thank you for your question. In the control groups, we intra-subicular injected 500 nL saline before kindling stimulation. To avoid misleading, we’ve directly indicated that the first column in the revised **Figure 7a and b** was saline line group.

Comment 30

“16. Page 14. “Both blockage of subicular HCN channels and optogenetic inhibition of the SUB-ANT circuit suppressed...” How optogenetic inhibition was achieved here?”

Response 30:

Thank you for your question. We apologize for unclear description in the method section. Optogenetic inhibition and ZD7288 injection here were conducted during the kindling acquisition process until the mice were fully kindled similarly done in the Figure 2i and j, we’ve indicated in the method section of the revised manuscript.

Page 15 paragraph 2 in the revised manuscript:

“Next, we tested the influence of blockage of subicular HCN channels or optogenetic inhibition of the SUB-ANT circuit during kindling acquisition on the strength of synaptic plasticity.”

Comment 31

“17. Figure 7e-i cannot not be published in its present form. The authors state that they have measured field potentials in ANT elicited by subicular stimulation. However, it appears that the tentative evoked potentials are traces chosen arbitrarily in the period after the stimulation artefact (Figure 7e). Further, the field potentials have a half-width of about 30 microseconds, which is biologically meaningless. Evoked potentials elicited by SUB stimulation should not be referred to as spikes to avoid confusion with single-unit (spike) terminology.”

Response 31:

Thank you very much for this important comment.

First, we apologize that the 2 last FP examples were not aligned with the trace in the original figure 7, which make it look like arbitrarily chosen. We have corrected mislabeling in our revised **Figure 7e**.

Meanwhile, we have carefully checked our original data of the sample traces, and the half-width of the FPs should be about 20 milliseconds (**Figure 8 for Reviewers**).

Finally, as suggested, to avoid any confusion, we have revised 'potential' instead of the term 'spike' in the last section of results.

Revised Figure 7e

Revised Figure 7g

Figure 8 for Reviewers

Comment 32

"18. Page 24. "Wildtype mice were injected with CTB-555..." What is CTB-555?"

Response 32:

Thank you for your question. CTB-555 refers to cholera toxin subunit B conjugated to Alexa-555, and we've added this explanation in the method section.

Comment 33

"19. In the discussion and figure, the phrasing "hyperexcitatory neurons" suggests the existence of a unique population of hyperexcitatory neurons in the subiculum. Subicular calcium signaling indeed increased during seizures but it is not demonstrated that these neurons are hyperexcitatory in their normal firing."

Response 33:

Thank you for your comments. We agreed that subicular pyramidal were not hyper-excitatory in physiological state and such hyper-excitatory condition was induced by epileptic seizures. As suggested, we've rephased the sentences as "Subicular pyramidal neurons control the generalization of hippocampal seizures".

Comment 34

"20. The authors suggest that subicular neurons gate the generalization of hippocampal seizures. However, all mice populations (Chr2, ArchT or control in SUB) progressed to at least stage 3 seizures with repeated kindling."

Response 34:

Thank you for your comments. We've rephrased the 'control' instead of the term 'gate' in the revised manuscript.

Minor Issues**Comment 35**

"1. Page 22 – "rearing and falling" instead of "rearing and failing""

Response 35:

Thank you. We've corrected the typo.

Comment 36

"2. Page 5 – promotor to promoter"

Response 36:

Thank you. We've corrected the typo.

Comment 37

"3. Page 5 – "...we first created non-specific lesions in the subiculum..." How lesions were done is not described in the method section."

Response 37:

Thank you. We've added descriptions about how lesions were done in the method section of the revised manuscript.

Page 24 paragraph 1 in the revised manuscript:

"For electrical lesion studies, we used 1 mA, 10 s, direct current stimulation in the subiculum".

Comment 38

"4. Page 4 - "Adopted an adeno associated virus... into the subiculum" is awkward phrasing."

Response 38:

Thank you. We've rephrased the whole sentence as "We first focused on functional changes of pyramidal neurons, the main type of neurons in the subiculum controlling neural transmission during hippocampal seizures. For this purpose, we expressed the genetically encoded calcium indicator GCaMP6s into subicular pyramidal neurons of wild-type mouse under the Ca/calmodulin dependent protein kinase II α (*CaMKII α*) promoter".

Comment 39

"5. Figure S7 change "glutamnergic" to "glutamatergic". Same in page 5"

Response 39:

Thank you. We've corrected the typo throughout the revised manuscript.

Comment 40

"6. Page 7 – change "optical cannula on the ANT" to "optical cannula in the ANT""

Response 40:

Thank you. We've rephrased the sentence as "We implanted an optical cannula in the ANT of CaMKII α -Chr2 SUB mouse".

Comment 41

"7. Figure 5g and 5h, indicate what the black bars represent."

Response 41:

Thank you. We've indicated that the black bars represented kindling stimulation period in the revised figure legend.

Comment 42

"8. Page 11 – "...monitored fluorescence changes during hippocampal seizures (Figure 5e and f)." No seizures are seen in these panels."

Response 42:

Thank you. We've added corresponding FS and GS samples in the revised **Figure 5**.

Revised Figure 5g-n

Comment 43

“9. Page 25 – What is your definition of fully kindled animals?”

Response 43:

Thank you for your question. Mice that had three successive stage 5 seizures after kindling induced seizures were regarded as fully kindled. We’ve added descriptions in “Single pulse measurement” section of the methods in the revised manuscript.

Comment 44

“10. For all ArchT experiments it would be more appropriate to use term hyperpolarization than inhibition.”

Response 44:

Thank you for this suggestion. We have used “hyperpolarization” instead of “inhibition” in all ArchT experiments in the revised manuscript.

Comment 45

“11. Page 15 – “...this effect was mediated by HCN channel-based bursting activities.” HCN can control the extent of bursting, but it does not mediate bursting. In brain neurons, the bursting is mediated by Ca2+ channels and possibly to some extent persistent Na current.”

Response 45:

Thank you. We’ve rephrased these sentences as “this effect was associated with HCN channel-contributed bursting activities”.

REVIEWER COMMENTS

Reviewer #1 (Remarks to the Author):

In the revised version, the authors addressed all my comments and performed the requested analysis and experiments. I fully support publication.

Reviewer #2 (Remarks to the Author):

Fei et al. have replied satisfactorily to most of my comments. However, there are still two issues that were not addressed properly.

- Their new results show that blocking T-type calcium current in subicular neurons has an antiseizure effect. This is in agreement with previous findings (PMID: 28326015) that the authors omitted to mention. This reference should be cited.

- Contribution of I_h to burst firing. By knocking down HCN1, the authors show that HCN contributes to the resting membrane potential of subicular pyramidal cells. Because the number of spikes per burst is reduced in KD animals, they conclude that HCN contributes to burst firing. However, the data presented are not convincing as the values of membrane potentials are not reported. T-type calcium current is voltage dependent and sensitive to the membrane potential that precedes depolarization due to its inactivation gating property (see PMID: 12506128). The authors should perform recordings while varying the membrane potential in a systematic manner and plot the number of spikes as a function of V_m.

Reviewer #3 (Remarks to the Author):

Thank you for your detailed responses to the review. The authors have conducted additional experiments to address the concerns raised in the initial review and have improved the manuscript. However, some major concerns outlined in the initial review were perhaps misunderstood and have not been fully addressed:

Comment 8. Please provide exact criteria for the detection of the end of the paroxysmal discharge: i.e. amplitude, threshold etc. It is important to provide a formal definition for the end of the paroxysmal

discharge otherwise, “durations” in after-discharge durations can vary arbitrarily from animal to animal and can include possible investigator bias.

Further, hippocampal theta varies in amplitude according to behavioural states. “Base” and “seizure” periods in these examples differ only in amplitude and could be due to spontaneous theta variability. Can authors validate these focal seizures by other methods?

Comment 16 was not addressed. We do not see anatomical landmarks despite the authors enhancing the background noise. Please provide histological slices with a larger field of view (i.e. we should see at least the hippocampus so we can judge whether injections were delivered to the right locations).

Comment 21 was not addressed.

Line 624: By definition, if you recorded multi-unit activity, it is not possible to say anything about pyramidal neurons i.e. single-units. The authors have only changed the wording from “single-unit” to “multi-unit” but have included “single-unit” analysis in Figure 6 and in the results.

New concerns:

Figure S1: Electrolytic lesion of the subiculum is not visible; we see traumatic injury of the cortex, commissural fibers and the subiculum from the implant but not the characteristic “burn” of the electrolytic lesion. Indicate location of subiculum. Considering the complete destruction of the subiculum, it is unclear where the subicular LFP is coming from.

Figure 2:

b and j: Control data look significantly different in j and b. Seizure stage progression from control data in j (CamKIIa-eGFP) is relatively equal to seizure progression for Chr2 data in b (if the two were compared, they would overlap). Similarly, for ADD data: if we were to compare the control data from b with ArchT data in j, they would overlap. In other words, if both control data from j and b are compared, they are likely to differ statistically. This calls into question all subsequent statistical comparisons.

REVIEWER COMMENTS

Reviewer #1 (Remarks to the Author):

General Comment

“In the revised version, the authors addressed all my comments and performed the requested analysis and experiments. I fully support publication.”

Response: We would like to express our sincere appreciation for your time to review our manuscript and the positive comments on this study.

Reviewer #2 (Remarks to the Author):

General Comments

“Fei et al. have replied satisfactorily to most of my comments. However, there are still two issues, that were not addressed properly.”

Response: We would like to express our sincere appreciation for your time to review our manuscript again and the constructive comments on this study, which are all valuable and very helpful for improving our paper. We hope that the following responses according to your comments could address your concerns. In the remainder of this letter, we discuss each of your comments individually along with our corresponding responses.

To facilitate this discussion, we first retype your comments in *italic font* and then present our responses to the comments in blue color.

Comment 1

“Their new results show that blocking T-type calcium current in subicular neurons has an antiseizure effect. This is in agreement with previous findings (PMID: 28326015) that the authors omitted to mention. This reference should be cited.”

Response 1:

Thank you very much for your suggestion. We’ve cited this pioneering work in the discussion part of the revised manuscript.

Page 19 paragraph 1 in the revised manuscript:

“Pharmacological blockage of T-type Ca^{2+} channel significantly inhibited bursting intensity of subicular pyramidal neurons and hippocampal seizures, which is in agreement with previous findings⁶³, further confirming that bursting firing in subicular pyramidal neuron is important in generalization of hippocampal seizures.”

Comment 2

“Contribution of Ih to burst firing. By knocking down HCN1, the authors show that HCN contributes to the resting membrane potential of subicular pyramidal cells. Because the number of spikes per burst is reduced in KD animals, they conclude that HCN contributes to burst firing. However, the data presented are not convincing as the values of membrane potentials are not reported. T-type calcium current is voltage dependent and sensitive to the membrane potential that precedes depolarization due to its inactivation gating property (see PMID: 12506128). The authors should perform recordings while varying the membrane potential in a systematic manner and plot the number of spikes as a function of Vm.”

Response 2:

Thank you very much for this important suggestion. Firstly, we have reported the values of membrane potential in the revised Figure 6k. Meanwhile, following your suggestion, we’ve have performed patch recordings while varying the membrane potential from -55 mV to -70 mV at

gradient of 5 mV and plot the number of spikes as a function of membrane potential in the revised Figure S12. Interestingly, we found that ANT-projecting subicular pyramidal neurons showed a tendency of decrease in AP numbers on burst when holding at more hyperpolarizing membrane potentials, suggesting that bursting firing may be much more sensitive to the depolarizing membrane potential. Further, knockdown of HCN1 in ANT-projecting subicular pyramidal neurons, which led to a slight hyperpolarization, reduced AP numbers on burst in each membrane potential (Figure S12). Our finding is consistent with previous studies that bursting in subicular neurons may occur at more depolarizing membrane potentials^{1,2}. This seems to be different from “de-inactivation” theory of T-type calcium current³. Classically, bursting firing is usually induced shortly after delivery of a hyperpolarizing pulse which let the T-type channels recover from inactivation, so-called “de-inactivation.” Thus, bursting can be triggered by neurons with the resting membrane potential hyperpolarized to -70 mV or greater. One possible explanation for our results here could be caused by the “window current” phenomenon of T-type calcium channels^{4,5}. There might be a small but obvious window component of calcium currents in the subiculum and only a small fraction of the T-type calcium channel population activation was enough to generate a robust bursting near membrane potential. Alternatively, bursting in the subiculum could also be driven primarily by non-inactivating, high-voltage-activated calcium tail currents by an action potential¹, distinguishing the burst mechanisms in the subiculum from those of thalamic neurons. Thus, we have included this new data in the revised **Figure S12** and added corresponding discussion in the revised manuscript as following:

Revised Figure S12

Reviewer #3 (Remarks to the Author):

General Comment

“Thank you for your detailed responses to the review. The authors have conducted additional experiments to address the concerns raised in the initial review and have improved the manuscript. However, some major concerns outlined in the initial review were perhaps misunderstood and have not been fully addressed.”

Response: We would like to express our sincere appreciation for your time to review our manuscript again and the valuable comments on this study, which are all very helpful for improving our paper. We hope that the following responses according to your comments could address your concerns. In the remainder of this letter, we discuss each of your comments individually along with our corresponding responses.

To facilitate this discussion, we first retype your comments in *italic font* and then present our responses to the comments in blue color.

Comment 1

“Comment 8. Please provide exact criteria for the detection of the end of the paroxysmal discharge: i.e. amplitude, threshold etc. It is important to provide a formal definition for the end of the paroxysmal discharge otherwise, “durations” in after-discharge durations can vary arbitrarily from animal to animal and can include possible investigator bias. Further, hippocampal theta varies in amplitude according to behavioural states. “Base” and “seizure” periods in these examples differ only in amplitude and could be due to spontaneous theta variability. Can authors validate these focal seizures by other methods?”

Response 1:

Thank you very much for these comments.

(1) For the criteria of ADD in kindling model, it was defined as the duration between the moment of kindling stimulation and the end of paroxysmal discharge event in EEG. These continuous discharges showed an average amplitude >3 times versus baseline. The end of ADD in our study was defined as the end of kindling-induced spiking activity, and isolated post-paroxysmal spikes were not calculated in the ADD. This criterion is according to previous study⁶. We've also marked the ending in sample eGFP-GS EEG as following (**Figure 1 for reviewers**). All behavioral assessment and EEG analysis were performed by well-trained experimenters blinded to the group allocation. To make it clearer, we have added detailed descriptions in the method section.

Figure 1 for reviewers. Red triangles indicated the end of ADD.

Page 24 paragraph 1 in the revised manuscript:

“The length of ADD was defined as the duration between the moment of kindling stimulation and the end of paroxysmal discharge event in EEG. These continuous discharges showed an average amplitude > 3 times versus baseline and isolated post-paroxysmal spikes were not calculated in the ADD according to previous study.”

(2) For the criteria of seizure in KA model, a FS is defined as a sharp paroxysmal event that continued more than 10 s and had an average amplitude > 3 times versus baseline and frequency > 2 Hz. Seizure analysis were performed by well-trained experimenters blinded to the group allocation. Actually, FS patterns in different animals indeed varied in their frequency and amplitude, but all these paroxysmal activities were characterized by sharp spike waves with unidirectional high amplitude (Samples 1-3, **Figure 2 for reviewers**), which is also reported in previous studies^{7,8}. This feature can be used to distinguish physiological theta activities in the hippocampus that are usually reported as bidirectional waves with low amplitude^{9,10}. To make it clearer, we have showed the enlarged view of typical EEGs in the revised **Figure 4c**.

Figure 2 for reviewers. Different types of FS patterns in the KA model.

Revised Figure 4c

Comment 2

“Comment 16 was not addressed. We do not see anatomical landmarks despite the authors enhancing the background noise. Please provide histological slices with a larger field of view (i.e. we should see at least the hippocampus so we can judge whether injections were delivered to the right locations).”

Response 2:

Thank you very much for this comment. We apologize for the misunderstanding and did not include histological slices of volume injection areas in the former revision. Accordingly, we have provided larger field of histological slices including hippocampal areas in the revised manuscript (please see details in the revised **Figures 2, 3, 4, 5, S3, S4, S8**).

Revised Figure 2a as a sample:

Revised Figures 2a

Comment 3

“Comment 21 was not addressed. Line 624: By definition, if you recorded multi-unit activity, it is not possible to say anything about pyramidal neurons i.e. single-units. The authors have only changed the wording from “single-unit” to “multi-unit” but have included “single-unit” analysis in Figure 6 and in the results.”

Response 3:

Thank you very much for your comments. We do agree with you that description about pyramidal neurons or using single-unit analysis was not appropriate. As you mentioned previously that analysis errors of spike sorting may come when neuron firing synchronously or in a bursting state, to avoid any misleading, we've deleted the results about pyramidal neurons in single-unit data during seizure state. To simplify, we only reserved multi-unit analysis in base state. The results show units in the SUBa and SUBb were not discriminative in their average firing rates, while units in the SUBa had a more robust bursting firing ability (higher number of bursts per min, **Figure 6b-d**). This is further verified by our CTB-based slice electrophysiology. These data indicated that ANT-projecting subicular pyramidal neurons, mainly distributed in the deep layer of the subiculum, showed enhanced bursting intensity. Please see in the **revised Figure 6**.

Revised Figure 6

Comment 4

“New concerns: Figure S1: Electrolytic lesion of the subiculum is not visible; we see traumatic injury of the cortex, commissural fibers and the subiculum from the implant but not the characteristic “burn” of the electrolytics lesion. Indicate location of subiculum. Considering the complete destruction of the subiculum, it is unclear where the subicular LFP is coming from.”

Response 4:

Thank you very much for this comment. The traumatic injury might result from the mechanical damage during electrode withdraw from the brain. This process is performed before perfusion of histology experiment. While, when the electrode is still in the brain of behaving mice, very flat LFPs could be recorded from the subiculum, which might be due to volume conduction from surround tissues^{11,12}. We hope this explanation could address your concern.

Alternatively, as electrolytic lesion is not specific as optogenetics in the main figures, to avoid any confusion, we would delete this part of results according to your further comment.

Comment 5

“Figure 2: b and j: Control data look significantly different in j and b. Seizure stage progression from control data in j (CamKIIa-eGFP) is relatively equal to seizure progression for Chr2 data

in b (if the two were compared, they would overlap). Similarly, for ADD data: if we were to compare the control data from b with ArchT data in j, they would overlap. In other words, if both control data from j and b are compared, they are likely to differ statistically. This calls into question all subsequent statistical comparisons.”

Response 5:

Thank you very much for this comment. Following your suggestion, we put two control groups from figures 2b and j in one figure and make a comparison, we found that there was no statistical significance between the control groups of figures 2b and j ($F(1,17) = 1.111, P = 0.3065$ for seizure stage; $F(1,17) = 1.607, P = 0.2221$ for ADD, Two-way RM ANOVA, **Figure 3 for reviewers**).

Although there is no statistical significance, we indeed see a minor difference between the control groups of figures 2b and j, which might be due to many factors like different color, frequency band used in light stimulation, the minor injury caused by the optical cannula et al. Thus, we used rigorous control in each independent experiment. All original data are provided in the **Raw Data file**.

Figure 3 for reviewers. Comparison of control groups in the Figures 2b and 2j.

References

- 1 Jung, H. Y., Staff, N. P. & Spruston, N. Action potential bursting in subicular pyramidal neurons is driven by a calcium tail current. *J Neurosci* 21, 3312-3321, doi:10.1523/jneurosci.21-10-03312.2001 (2001).
- 2 Cooper, D. C., Chung, S. & Spruston, N. Output-mode transitions are controlled by prolonged inactivation of sodium channels in pyramidal neurons of subiculum. *PLoS biology* 3, e175, doi:10.1371/journal.pbio.0030175 (2005).
- 3 Cheong, E. & Shin, H. S. T-type Ca²⁺ channels in normal and abnormal brain functions. *Physiol Rev* 93, 961-992, doi:10.1152/physrev.00010.2012 (2013).
- 4 Crunelli, V., Tóth, T. I., Cope, D. W., Blethyn, K. & Hughes, S. W. The 'window' T-type calcium current in brain dynamics of different behavioural states. *J Physiol* 562, 121-129, doi:10.1113/jphysiol.2004.076273 (2005).
- 5 Dreyfus, F. M. *et al.* Selective T-type calcium channel block in thalamic neurons reveals channel redundancy and physiological impact of I(T)window. *J Neurosci* 30, 99-109, doi:10.1523/jneurosci.4305-09.2010 (2010).
- 6 Racine, R. J. Modification of seizure activity by electrical stimulation. II. Motor seizure. *Electroencephalography and clinical neurophysiology* 32, 281-294, doi:10.1016/0013-4694(72)90177-0 (1972).
- 7 Bouilleret, V. *et al.* Recurrent seizures and hippocampal sclerosis following intrahippocampal kainate injection in adult mice: electroencephalography, histopathology and synaptic reorganization similar to mesial temporal lobe epilepsy. *Neuroscience* 89, 717-729, doi:10.1016/s0306-4522(98)00401-1 (1999).
- 8 Klein, S., Bankstahl, M. & Löscher, W. Inter-individual variation in the effect of antiepileptic drugs in the intrahippocampal kainate model of mesial temporal lobe epilepsy in mice. *Neuropharmacology* 90, 53-62, doi:10.1016/j.neuropharm.2014.11.008 (2015).
- 9 Itskov, V., Pastalkova, E., Mizuseki, K., Buzsáki, G. & Harris, K. D. Theta-mediated dynamics of spatial information in hippocampus. *J Neurosci* 28, 5959-5964, doi:10.1523/jneurosci.5262-07.2008 (2008).
- 10 Patel, J., Fujisawa, S., Berényi, A., Royer, S. & Buzsáki, G. Traveling theta waves along the entire septotemporal axis of the hippocampus. *Neuron* 75, 410-417, doi:10.1016/j.neuron.2012.07.015 (2012).
- 11 Lagerlund, T. D. J. C. N. S. Volume conduction. 66, 28-40 (2002).
- 12 van den Broek, S. P., Reinders, F., Donderwinkel, M. & Peters, M. J. Volume conduction effects in EEG and MEG. *Electroencephalography and clinical neurophysiology* 106, 522-534, doi:10.1016/s0013-4694(97)00147-8 (1998).

REVIEWERS' COMMENTS

Reviewer #2 (Remarks to the Author):

The authors have replied satisfactorily to all my comments.

Reviewer #3 (Remarks to the Author):

The authors satisfactory responded to my critique. I have no further comments and support publication.

Reviewer #2 (Remarks to the Author):

The authors have replied satisfactorily to all my comments.

Response: We would like to express our sincere appreciation for your time to review our manuscript again and the positive comments on this study.

Reviewer #3 (Remarks to the Author):

The authors satisfactory responded to my critique. I have no further comments and support publication.

Response: We would like to express our sincere appreciation for your time to review our manuscript again and the support for publication.